# CyclicReflex: Improving Reasoning Models via Cyclical Reflection Token Scheduling

**Chongyu Fan**[†]  **Yihua Zhang**[†]  **Jinghan Jia**[†]  **Alfred Hero**[‡]  **Sijia Liu**[†,§]

[†]Michigan State University
[‡]University of Michigan, Ann Arbor
[§]IBM Research

## Abstract

Large reasoning models (LRMs), such as OpenAI's o1 and DeepSeek-R1, harness test-time scaling to perform multi-step reasoning for complex problem-solving. This reasoning process, executed before producing final answers, is often guided by special juncture tokens that prompt self-evaluative reflection. These transition markers and reflective cues are referred to as *"reflection tokens"* (*e.g.*, "wait", "but", "alternatively"). In this work, we treat reflection tokens as a *"resource"* and introduce the problem of *resource allocation*, aimed at improving the test-time compute performance of LRMs by adaptively regulating the frequency and placement of reflection tokens. Through empirical analysis, we show that both excessive and insufficient use of reflection tokens, referred to as over-reflection and under-reflection, can degrade model performance. To better understand this trade-off, we draw an analogy between reflection token usage and learning rate scheduling in optimization. Building on this insight, We propose *cyclical reflection token scheduling* (termed CyclicReflex), a training-free decoding strategy that dynamically modulates reflection token logits with a bidirectional, position-dependent triangular waveform, incurring no additional computation cost. Experiments on MATH500, AIME2024/2025, AMC2023, GPQA Diamond and LiveCodeBench demonstrate that CyclicReflex consistently improves performance across model sizes (1.5B–14B), outperforming standard decoding and recent approaches such as TIP (thought switching penalty) and S1. Codes are available at `https://github.com/OPTML-Group/CyclicReflex`.

## 1 Introduction

There has been a recent surge in the development of large reasoning models (LRMs), driven by the introduction of chain-of-thought (CoT) (Wei et al., 2022). Notable examples include OpenAI's o1 (OpenAI, 2024), Qwen 2.5 (Yang et al., 2024), DeepSeek-R1 (Guo et al., 2025), and Kimi-1.5 (Team et al., 2025). These models perform multi-step reasoning by generating so-called reflection tokens, phrases such as *"wait", "but", "alternatively"*, which signal hesitation, alternative exploration, or intermediate analysis. In parallel, test-time scaling (Snell et al., 2024; Liu et al., 2025; Sadhukhan et al., 2025) have emerged as a complementary approach for improving reasoning accuracy by expanding the breadth or depth of CoT traces during inference.

However, LRMs remain prone to reasoning failures due to mismanagement of reflection tokens, often resulting in either *underthinking* or *overthinking*. Underthinking occurs when the model fails to fully explore promising reasoning paths for complex problems, often terminating prematurely or switching strategies too soon (Wang et al., 2025b; Su et al., 2025). In contrast, overthinking arises when the model generates an excessive number of reflection tokens on simple problems, leading to unnecessary computational overhead (Chen et al., 2024; Kumar et al., 2025a). These observations show that, as internal signals for deliberative reasoning, reflection tokens play a critical role in shaping answer quality. The emerging challenges further underscore the need for a principled mechanism to regulate reflection token usage during inference.

In this work, we introduce the concept of **resource allocation for LRMs**, treating reflection tokens as a valuable resource whose scheduling along the CoT trajectory (*a.k.a.* reasoning trace) can be

strategically designed to improve reasoning accuracy. The objective is to optimize the quantity and placement of reflection tokens, adapting dynamically to the reasoning schedule and the difficulty of the current problem. For instance, some problems exhibit *under-reflection*, where too few reflection tokens result in premature answer generation, while others suffer from *over-reflection*, where excessive tokens stall progress by repeatedly looping on phrases like "wait". This raises a central question:

> **(Q)** *How can we achieve effective resource allocation in LRMs to mitigate both under-reflection and over-reflection?*

To answer this, **we draw a conceptual analogy between reflection tokens in LRMs and learning rates in optimization**. Leveraging the landscape of thoughts (Zhou et al., 2025), we show that under-reflection mirrors the effect of an overly small learning rate, leading to convergence to suboptimal solutions, while over-reflection resembles a large learning rate that causes divergence. We briefly introduce our motivation and the underlying intuition below.

**Overview of motivation and rationale: From stepsize hedging to cyclical learning rates.** The critical role of learning rates (also known as stepsizes) in shaping optimization dynamics has been extensively studied (Nesterov, 1983; Allen-Zhu & Orecchia, 2014; Bubeck et al., 2015). A recent theoretical advancement, the *silver stepsize schedule* (Altschuler & Parrilo, 2024; 2025), demonstrates that replacing a constant learning rate with an approximately periodic, *hedging-style* schedule can *provably accelerate convergence* in gradient descent. This approach is known as *stepsize hedging* as it alternates strategically between large and small stepsizes, balancing rapid (but potentially unstable) exploration with slower, more stable convergence. A similar stepsize hedging idea has been applied to deep model training through *cyclical learning rate schedule* (Smith, 2017), which alternates between large and small learning rates in a triangular waveform. This strategy not only accelerates convergence but also enhances generalization, often eliminating the need for extensive hyperparameter tuning.

Motivated by the principle of stepsize hedging (Altschuler & Parrilo, 2024; 2025) and the demonstrated effectiveness of cyclical learning rates in deep learning (Smith, 2017), we propose **CyclicReflex**, a training-free decoding strategy that dynamically modulates the logits of reflection tokens using a position-dependent, periodic triangular waveform (see **Fig. 1**). Just as cyclical learning rates alternate between aggressive and conservative updates to balance exploration and convergence, CyclicReflex cyclically adjusts the sampling likelihood of reflection tokens to regulate the depth and stability of the reasoning process. Unlike conventional decoding methods, CyclicReflex is *bidirectional*, capable of both promoting and suppressing reflection token usage depending on the stage of generation. This flexibility enables CyclicReflex to address both under-reflection (insufficient reasoning) and over-reflection (excessive, redundant reasoning), offering a principled mechanism for reasoning modulation inspired by optimization dynamics.

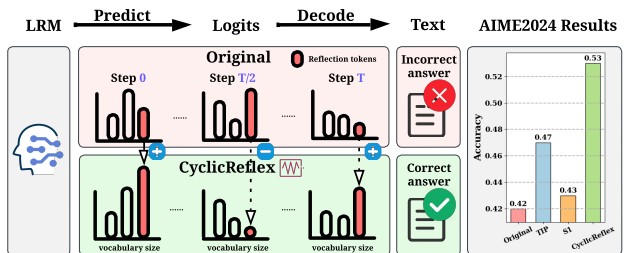

Figure 1: Schematic overview of our proposed method (CyclicReflex). The rightmost subfigure presents a comparison of final answer accuracy between CyclicReflex, the original LRM (DeepSeek-R1-Distill-Llama-8B), and decoding variants using TIP (Wang et al., 2025b) and S1 (Muennighoff et al., 2025).

**Contributions.** We summarize our contributions below.

• We introduce and formalize the problem of resource allocation in LRMs by treating reflection tokens as a computational resource, motivated by the dual challenges of under-reflection and over-reflection in reasoning generation.

• We draw a conceptual analogy between reflection token scheduling and learning rate scheduling in optimization, and validate it through the landscape of thoughts. Guided by that, we propose CyclicReflex, a test-time decoding strategy that cyclically modulates reflection token logits to dynamically balance reflection during generation.

• We conduct comprehensive experiments across six reasoning benchmarks and multiple model scales (1.5B–8B), demonstrating that CyclicReflex consistently improves both final-answer accuracy and self-correction capability, outperforming recent methods such as TIP (Wang et al., 2025b) and S1

(Muennighoff et al., 2025). Moreover, CyclicReflex integrates seamlessly with other test-time scaling techniques, yielding additional performance gains.

## 2 RELATED WORK

**LRMs and CoT.** CoT (Wei et al., 2022) enables LRMs to solve complex tasks through intermediate reasoning steps before reaching a final answer. This technique underpins many recent LRMs, including OpenAI's o1 (OpenAI, 2024), Qwen 2.5 (Yang et al., 2024), DeepSeek-R1 (Guo et al., 2025), and Kimi-1.5 (Team et al., 2025), which often employ reinforcement learning to further improve their reasoning performance. Guo et al. (2025) show that even smaller models benefit substantially from fine-tuning with CoT-style data. A hallmark of CoT reasoning in these models is the emergence of reflection tokens (words like "wait" or "but") that signal deliberation or self-correction, marking a shift from fast to slow thinking (Kumar et al., 2025b; Li et al., 2025). In this paper, we show that the reasoning performance of LRMs can be enhanced by applying cyclical logits manipulation to reflection tokens.

**Efficient reasoning.** Despite their impressive capabilities, LRMs often exhibit reasoning inefficiencies. Overthinking arises when the model generates unnecessarily long reasoning traces, leading to inflated outputs and increased computational cost (Chen et al., 2024; Kumar et al., 2025a). In contrast, underthinking occurs when the model halts reasoning too early, failing to adequately explore promising solution paths (Wang et al., 2025b; Su et al., 2025). Therefore, ensuring both the efficacy (*i.e.*, answer accuracy) and efficiency (*i.e.*, generation length) of reasoning is crucial. Building on this line of research, some approaches modify model behavior through post-training interventions. For instance, Luo et al. (2025); Aggarwal & Welleck (2025); Hou et al. (2025) use fine-tuning or reinforcement learning to explicitly control reasoning length. There also exist works that adopt training-free strategies. Yang et al. (2025c) propose guiding smaller models with larger ones at inference; Wang et al. (2025b) penalize reflection token logits to reduce over-reflection; And Yang et al. (2025b); Muennighoff et al. (2025) develop early-exit mechanisms for efficient decoding. Our method also falls into the training-free category but differs in its dynamic to adaptively address both under- and over-reflection without model modification.

**Test-time scaling.** A growing body of work enhances LRM reasoning via test-time scaling. Basic strategies include manually inserting reflection tokens (*e.g.*, "wait", "but") to prompt deeper thinking (Muennighoff et al., 2025; Jin et al., 2025; Chen et al., 2025c). More methods such as Best-of-N generation and self-consistency sampling (Wang et al., 2022; Irvine et al., 2023; Brown et al., 2024; Kang et al., 2025; Wang et al., 2025a) aim to select the most promising answer among multiple candidates, often guided by reward models. Structured decoding approaches, such as beam search (Feng et al., 2023), tree-of-thought (ToT) (Yao et al., 2023), and Monte Carlo tree search (MCTS) (Zhou et al., 2023), further improve answer quality by enabling the model to reason over multiple candidate paths. In pursuit of controlled reasoning, Wu et al. (2025) propose thinking intervention, which selectively inserts or edits specific thinking tokens during generation to tailor LRM behavior for downstream tasks. Recent analyses (Snell et al., 2024; Liu et al., 2025; Chen et al., 2025b; Zhang et al., 2025; Ghosal et al., 2025) also highlight that the effectiveness of test-time scaling varies with problem difficulty, motivating strategies that adapt to instance complexity. Our method provides the adaptive test-time compute by dynamically adjusting the influence of reflection tokens throughout the reasoning trajectory. We show that it consistently improves accuracy across difficulty levels and can be integrated seamlessly with other test-time strategies like Best-of-N and beam search.

## 3 RESOURCE ALLOCATION IN REASONING MODELS

In this section, we begin by introducing preliminaries on LRMs, including their chain-of-thought trajectories (*i.e.*, reasoning traces) and the use of reflection tokens. We then motivate the problem of resource allocation over reflection tokens through an existing technique: thought switching penalty (TIP) test-time compute strategy. This warm-up study illustrates the critical influence and sensitivity of reflection tokens on the final answers produced by LRMs.

**Preliminaries on LRMs, reasoning traces, and reflection tokens.** Unlike conventional LLMs, LRMs can incorporate an explicit *thinking stage* before arriving at a final answer (Li et al., 2025;

Kumar et al., 2025b; Chen et al., 2025a). This thinking stage is typically realized through a CoT trajectory composed of multiple reasoning steps.

To be concrete, let $\mathbf{x}$ denote the input question, $\mathbf{r}$ the reasoning trace, and $\mathbf{y}$ the final answer. The reasoning trace can be written as $\mathbf{r} = [\mathbf{r}_1, \mathbf{r}_2, \ldots, \mathbf{r}_T]$, a sequence of $T$ intermediate reasoning steps (or thoughts) that serve as a precondition for producing the final answer $\mathbf{y}$. These thoughts are often prefixed by lexical cues such as "*wait*", "*but*", and "*alternatively*". In this work, we collectively refer to such words as **reflection tokens**, as they signal the LRM's introspective or deliberative thinking during generation. These tokens typically signal hesitation, reconsideration or alternative exploration within the reasoning trace, serving as implicit markers of the model's engagement in step-by-step reasoning. For ease of understanding, **Table 1** illustrates an example where an LRM tackles a math problem ($\mathbf{x}$), along with its generated reasoning trace ($\mathbf{r}$), final answer ($\mathbf{y}$), and the highlighted reflection tokens embedded within the reasoning process.

Table 1: An example from MATH500 answered by DeepSeek-R1-Distill-Qwen-7B. Reflection tokens are highlighted in red.

| | Generation (r and y) vs. input x |
|---|---|
| **x** | Compute $\arcsin\left(-\frac{1}{2}\right)$. Express your answer in radians. |
| **r** | **\<think\>**Okay, so I need to compute arcsin of -1/2...... But I should make sure I'm thinking about the right range for arcsin. I think it's between $-\pi/2$ and $\pi/2$, right? So it's the principal value...... I know that sine is negative in the third and fourth quadrants, but since arcsin only gives me angles in $[-\pi/2, \pi/2]$, that's the fourth quadrant...... Wait, let me double-check. If I take $\sin(-\pi/6)$, that's the same as $-\sin(\pi/6)$, which is -1/2. So the answer should be $-\pi/6$ radians.**\</think\>** |
| **y** | The arcsin of -1/2 is $-\pi/6$ radians. |

**Motivation: Influence of reflection tokens on final answer accuracy.** Next, we present a *warm-up* study to illustrate the critical role of reflection tokens in determining final answer quality, particularly across varying levels of problem difficulty. Prior work has identified several common issues in LRM, including "underthinking" (Wang et al., 2025b; Su et al., 2025)–where the model prematurely abandons promising lines of thought, resulting in insufficient reasoning depth–and "overthinking" (Muennighoff et al., 2025; Chen et al., 2024; Kumar et al., 2025a)–where excessive, unnecessary reasoning steps obscure or derail the correct final answer. These studies have observed that intervening in the reasoning trace with reflection tokens can help guide the model's thought process (Wu et al., 2025). A common approach to controlling and integrating the effect of reflection tokens in LRM generation is to modify the decoding strategy to account for their occurrence. One such method is TIP (Wang et al., 2025b), which was proposed to discourage the generation of reflection tokens and thereby penalize frequent thought switches during the reasoning trace.

Given the set of reflection tokens $\hat{V}$, TIP introduces a logit penalty ($\alpha$) to the predicted score $z_{t,v}$ when generating a reflection token $v \in \hat{V}$ at reasoning step $t$, yielding the updated logit

$$\text{TIP}(\alpha): \quad \hat{z}_{t,v} = \begin{cases} z_{t,v} + \alpha, & \text{if } v \in \hat{V} \text{ and } t < T_0 \\ z_{t,v}, & \text{otherwise} \end{cases} \tag{1}$$

where $\alpha$ controls the strength of the logit intervention, and $T_0$ specifies the time window over which the adjustment is applied. For clarity, $\text{TIP}(\alpha)$ denotes the TIP-based decoding strategy parameterized by $\alpha$. It is worth noting that *TIP sets $\alpha \leq 0$ to penalize frequent thought switches.*

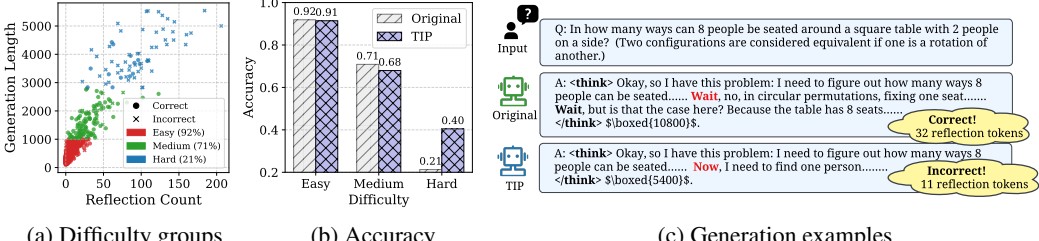

(a) Difficulty groups      (b) Accuracy      (c) Generation examples

Figure 2: (a) Answers from DeepSeek-R1-Distill-Qwen-7B on MATH500 clustered into Easy, Medium, and Hard using K-means over reflection word count and generation length. Each point represents one answer. (b) Accuracy of original decoding and TIP across difficulty levels. (c) Generation examples of original decoding and TIP for a problem from the Medium category.

Despite the heuristic nature of choosing $\alpha$ and $T_0$, the TIP-based decoding strategy (Wang et al., 2025b) provides useful motivation for studying resource allocation over reflection tokens and its impact on reasoning effectiveness (*e.g.*, on the MATH500) across *different problem difficulty levels*. As shown in **Fig. 2(a)**, we categorize MATH500 problems into three difficulty levels—Easy, Medium, and Hard—based on generation length and reflection token count, with the final-answer accuracies

92%, 71%, and 21%, respectively. We observe that for more challenging problems, LRMs tend to produce longer reasoning trajectories and more reflection tokens, indicating deeper engagement in problem-solving. **Fig. 2(b)** next compares the accuracy of the original decoding strategy and TIP across the difficulty groups in Fig. 2(a). As shown, TIP improves accuracy on Hard problems *but reduces accuracy on Easy and Medium problems*. This suggests that TIP's constant logit manipulation strategy (agnostic to the reasoning step $t$) does not yield *optimal* reasoning control. Furthermore, **Fig. 2(c)** shows a Medium-level example comparing original decoding with TIP. The first divergence in reasoning is highlighted in red. Under original decoding, the model introduces a transitional reflection ("Wait, no, in circular permutations, fixing one seat..."), generating 32 reflection tokens before arriving at the correct answer. In contrast, with a thought-switching penalty $\alpha < 0$ in (1), TIP reshapes the trace ("Now, I need to find one person..."), producing only 11 reflection tokens and yielding an incorrect answer. This shows that TIP provides only one-directional reflection control (penalizing reflection token logits). Hence, a bi-directional, dynamically adaptive (non-constant) reflection token allocation strategy is needed.

**Problem of interest: Resource allocation over reflection tokens.** Reflection tokens have a significant impact on the reasoning capability of LRMs. Therefore, if we view reflection tokens as a "resource" in LRM reasoning generation, then determining their schedule, including the number of occurrences and their positions, naturally gives rise to the *problem of resource allocation* for LRMs.

TIP offers a simple solution by applying a constant logit penalty to reflection token generation. However, this approach is *static* and therefore fails to account for both the number and placement of reflection tokens, which are dynamically determined during reasoning trace generation. As shown in Fig. 2(b), TIP does not consistently improve performance across all difficulty levels. This leaves open the question of how to schedule reflection token generation along the reasoning trajectory, that is, how to allocate these "resources" effectively over time while accounting for problem difficulty. These underscore the need for more adaptive and fine-grained strategies to control reflection token usage in order to address the resource allocation more effectively.

## 4 REFLECTION TOKEN SCHEDULING AS LEARNING RATE SCHEDULING IN OPTIMIZATION

In this section, we draw a conceptual analogy between reflection token scheduling and learning rate scheduling in optimization, aimed at deepening our understanding of reflection tokens in reasoning and enabling more effective resource allocation. And we propose a new decoding strategy: cyclical reflection token scheduling (**CyclicReflex**).

**Reflection tokens in the thought landscape vs. learning rates in the optimization landscape.** The role of reflection tokens in reasoning closely mirrors that of learning rates in optimization. In the "thought landscape", a model initiates by interpreting a question and leverages reflection tokens to modulate its reasoning trajectory: exploring, reconsidering, and refining intermediate steps before reaching a final answer. Likewise, in the optimization landscape, an optimizer begins from a random initialization and relies on the learning rate to control the step size of the variable updates, gradually converging toward an optimal solution. In both cases, a well-tuned control mechanism, reflection tokens in reasoning or learning rates in optimization, is essential for accurate solution convergence.

Additionally, in optimization, an improperly tuned learning rate, either too small or too large, can hinder convergence, causing the optimizer to either stagnate or diverge. This challenge in scheduling the learning rate maps onto the difficulty of scheduling reflection tokens in reasoning, manifesting as under-reflection and over-reflection.

(*Under-reflection*) When the model generates too few reflection tokens, it often terminates the reasoning process prematurely, resulting in a final answer that lacks sufficient deliberation. This is analogous to optimization with a learning rate that is too small, where the model converges too early and becomes trapped in a suboptimal local minimum.

(*Over-reflection*) Conversely, generating too many reflection tokens can prevent the model from concluding its reasoning, causing it to loop or stall, *e.g.*, repeatedly producing phrases like "wait" without reaching a solution. This resembles optimization with an overly large learning rate, which leads to instability and divergence rather than convergence.

To validate the analogy between reflection tokens and learning rates (too small learning rate vs. under-reflection, and too large learning rate vs. over-reflection), we utilize the interpretability tool introduced in (Zhou et al., 2025) to visualize the *landscape of thoughts*. This tool projects reasoning step $\mathbf{r}_i$ into a two-dimensional visual space based on the measured "distance" between each step $\mathbf{r}_i$ and the final answer $\mathbf{y}$, providing an interpretable view of the model's reasoning dynamics. The distance metric captures the model's uncertainty by taking the inverse of the probability of generating the answer $\mathbf{y}$ conditioned on the reasoning step $\mathbf{r}_i$, normalized by the length of $\mathbf{y}$:

$$d(\mathbf{r}_i, \mathbf{y}) = p_{\text{LRM}}(\mathbf{y} \mid \mathbf{r}_i)^{-1/|\mathbf{y}|}, \tag{2}$$

where $p_{\text{LRM}}$ denotes the prediction probability assigned by the LRM to the answer $\mathbf{y}$ given the reasoning step $\mathbf{r}_i$, and $|\mathbf{y}|$ denotes the length of $\mathbf{y}$.

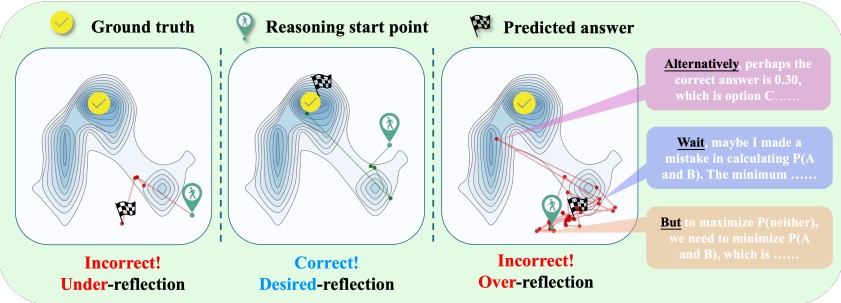

Figure 3: Examples of landscape of thought for under-reflection, desired-reflection, and over-reflection, generated by DeepSeek-R1-Distill-Qwen-7B with the original decoding strategy. Each point represents a reasoning step and is connected in the order of generation. Darker regions indicate steps with higher semantic alignment to the correct answer.

**Fig. 3** presents a visualized reasoning trajectory from the initial thought to the final answer under original decoding strategy, across three different scenarios: (i) under-reflection, where too few reflection tokens lead to a reasoning trace that is too short and results in an incorrect answer; (ii) desired reflection, which yields a well-structured reasoning trace and a correct answer; and (iii) over-reflection, where excessive reflection tokens cause an overly long and off-track reasoning trace, also resulting in an incorrect answer. In the landscape, darker regions represent intermediate reasoning steps that are semantically closer to the correct answer. That is, color intensity reflects the relative correctness of each thought along the trajectory.

As we can see, the thought landscape under under-reflection is too conservative to drive the reasoning process away from the starting point, ultimately failing to converge to the correct final answer. In contrast, over-reflection could enable the model to reach semantically promising regions of the landscape, for example, a step like "*Alternatively, perhaps the correct answer is ...*", which is far away from the thinking start point and located in the darker region. However, much like an excessively large learning rate that fails to properly control the optimization process, this leads the model to quickly pass through the desirable state without settling there, ultimately leading to an incorrect answer region. Moreover, we find that reflection tokens are responsible for the *sharp turns* in the reasoning trajectory. By examining the sharply turning steps in over-reflection, we observe that they are consistently initiated by reflection tokens.

**CyclicReflex: Cyclical logits manipulation for reflection token scheduling.** Although reflection tokens are crucial for guiding multi-step reasoning, balancing their use remains challenging. The need for dynamic modulation of reflection tokens closely mirrors the challenge of learning rate scheduling in optimization. As introduced in Sec. 1, the convergence of gradient descent can be *provably accelerated* by adopting the *silver stepsize schedule*, which follows the principle of *stepsize hedging* (Altschuler & Parrilo, 2024; 2025). The key algorithmic insight is to hedge between two individually suboptimal strategies, small and large stepsizes, since the failure modes of one are often mitigated by the strengths of the other. In deep model training, cyclical learning rates (Smith, 2017) exemplify this principle in practice. Rather than using a fixed learning rate, they employ a triangular waveform to periodically alternate between large and small step sizes. This schedule allows the optimizer to balance global exploration (enabled by large steps) with local convergence stability (provided by smaller steps), thereby yielding a form of stepsize hedging.

Inspired by cyclical learning rates, we introduce **CyclicReflex**. As depicted in **Fig. 4**, we apply a periodic triangular waveform to modulate the logits of reflection tokens during generation. The waveform is governed by two parameters: the amplitude $A$, which controls the strength of the logit adjustment, and the period $C$, which determines the oscillation frequency. This logits manipulation evolves over time and is both *position-dependent* (varying with each decoding step) and *bidirectional* (allowing for the dynamic promotion or suppression of reflection token sampling based on the current stage of the reasoning process). More concretely, CyclicReflex can be cast as the following logits manipulation as the function of the reasoning step $t$

$$\text{CyclicReflex}: \ \hat{z}_{t,v} = \begin{cases} z_{t,v} + \delta(t) & \text{if } v \in \hat{V}, \\ z_{t,v} & \text{otherwise}, \end{cases} \qquad \delta(t) = A \left| 4 \cdot \frac{(t - \frac{C}{4}) \bmod C}{C} - 2 \right| - A \quad (3)$$

where recall that the amplitude $A$ and the period $C$ have been previously defined as shown in Fig. 4, mod is the modulo operation, $|\cdot|$ is the absolute value operation, and the other notations follow (1). In (3), $(t - \frac{C}{4}) \bmod C$ gives the current thought position within the cycle, and it is straightforward to validate that $\delta(C/4) = A$ and $\delta(3C/4) = -A$.

As shown in Fig. 4, *CyclicReflex adopts a representative form of hedging schedule*: the increasing phase of the reflection logit adjustment $\delta(t)$ promotes exploration by encouraging the model to transition away from its current line of thought, while the decreasing phase fosters convergence by stabilizing the reasoning process, guiding the model toward producing a coherent final answer.

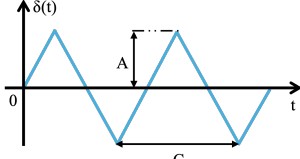

Compared to TIP (1), which applies a fixed unidirectional penalty, CyclicReflex adaptively modulates reflection token logits ~~with~~ at no additional computation cost, offering finer control over reasoning. This unified mechanism balances under- and over-reflection, yielding more robust and flexible behavior that adapts to the model's evolving thought process. In **Fig. A1** of **Appendix B**, we summarize the sensitivity of CyclicReflex to its hyperparameters. The period $C$ has a larger impact than the amplitude $A$.

Figure 4: Illustration of CyclicReflex (3), where $t$ denotes the token position and $\delta(t)$ the logit adjustment on reflection tokens, oscillating between $-A$ and $A$ with amplitude $A$ and period $C$.

## 5 EXPERIMENTS

### 5.1 EXPERIMENTAL SETUP

**Data-model setups.** To evaluate the effectiveness of CyclicReflex, we consider both *math* and *non-math* benchmarks. The math datasets include **MATH500** (Lightman et al., 2023) with 500 multi-step problems, **AIME2024/2025** (MAA Committees) with 30 challenging problems each year, and **AMC2023** (AI-MO, 2024) covering diverse competition topics. The non-math datasets include **GPQA Diamond** (Rein et al., 2024), a challenging subset of multiple-choice science questions in biology, chemistry, and physics, and **LiveCodeBench** (Jain et al., 2024), a coding benchmark from LeetCode, AtCoder, and Codeforces that evaluates code generation, repair, and execution. We evaluate our method using DeepSeek-R1-Distilled-Qwen (1.5B, 7B) (Guo et al., 2025) and DeepSeek-R1-Distilled-Llama-8B to assess different backbones. Additionally, we demonstrate generalization on Qwen3 (4B, 8B, 14B) (Yang et al., 2025a) in **Table A1** of **Appendix B**.

**Baseline and evaluation.** Our method (CyclicReflex) is compared against two primary baselines: **TIP** (Wang et al., 2025b), **S1** (Muennighoff et al., 2025). In addition, we assess the compatibility of CyclicReflex with external test-time scaling techniques, including **Best-of-N** (Irvine et al., 2023; Brown et al., 2024) and **Beam Search** (Feng et al., 2023), using RLHFlow-PRM-Deepseek-8B as the preference reward model (PRM) for scoring (Dong et al., 2024). We use **accuracy** and **generation length** as our primary evaluation metrics. Accuracy is obtained by rule-based extraction of the final answer against the ground truth, while generation length is the total word count of the response. More implementation details are provided in Appendix A.

### 5.2 EXPERIMENT RESULTS

**Overall Performance of CyclicReflex on the MATH Task.** In **Table 2**, we show the effectiveness of

Table 2: Accuracy (Acc) and generation length (Len) comparison on four math reasoning benchmarks (MATH500, AIME2024, AIME2025, and AMC2023) using DeepSeek-R1-Distilled Model: Qwen 1.5B, Qwen 7B, and Llama 8B. Each model is evaluated under four decoding strategies: *Original*, *TIP*, *S1*, and *CyclicReflex*. The best accuracy in each setting is highlighted in **bold**, while the second-best is underlined.

| Method | MATH500 | | AIME2024 | | AIME2025 | | AMC2023 | |
|---|---|---|---|---|---|---|---|---|
| | Acc | Len | Acc | Len | Acc | Len | Acc | Len |
| **DeepSeek-R1-Distill-Qwen-1.5B** | | | | | | | | |
| Original | 0.74 | 1253.05 | 0.23 | 3584.36 | 0.19 | 3442.07 | 0.63 | 1855.85 |
| TIP | 0.75 | 1206.91 | 0.23 | 3329.17 | 0.20 | 3825.17 | 0.63 | 1890.35 |
| S1 | 0.73 | 1532.05 | 0.17 | 4112.07 | 0.20 | 3867.71 | 0.45 | 3263.75 |
| CyclicReflex | **0.77** | 1212.94 | **0.30** | 3547.10 | **0.23** | 3467.97 | **0.65** | 1839.23 |
| **DeepSeek-R1-Distill-Qwen-7B** | | | | | | | | |
| Original | 0.86 | 785.25 | 0.43 | 2878.39 | 0.31 | 3192.59 | 0.81 | 1300.53 |
| TIP | 0.87 | 775.77 | 0.43 | 2806.53 | 0.30 | 3107.30 | 0.85 | 1267.83 |
| S1 | 0.83 | 1190.96 | 0.33 | 3541.10 | 0.33 | 3455.33 | 0.85 | 2158.00 |
| CyclicReflex | **0.89** | 777.93 | **0.50** | 2868.30 | **0.37** | 3190.33 | **0.90** | 1229.25 |
| **DeepSeek-R1-Distill-Llama-8B** | | | | | | | | |
| Original | 0.83 | 1196.98 | 0.42 | 3593.73 | 0.30 | 3922.41 | 0.81 | 1951.88 |
| TIP | 0.83 | 1080.62 | 0.47 | 3572.40 | 0.27 | 3866.00 | 0.85 | 1932.63 |
| S1 | 0.78 | 1461.93 | 0.43 | 3742.27 | 0.27 | 4351.87 | 0.75 | 2812.75 |
| CyclicReflex | **0.85** | 1108.30 | **0.53** | 3454.97 | **0.37** | 3856.80 | **0.90** | 1942.40 |

CyclicReflex across models of varying sizes (1.5B, 7B, and 8B), model families (Qwen and LLaMA), and four widely used reasoning benchmarks: MATH500, AIME2024, AIME2025, and AMC2023. As we can see, CyclicReflex consistently improves performance over the original LRM decoding strategy across all models and datasets. For example, DeepSeek-R1-Distill-Llama-8B with CyclicReflex achieves up to a 10% absolute accuracy gain on AIME2024, while DeepSeek-R1-Distill-Qwen 7B with CyclicReflex yields up to a 9% improvement on AMC2023. Additionally, these accuracy gains are achieved without sacrificing the efficiency of reasoning generation: CyclicReflex produces comparable reasoning traces relative to the original decoding method.

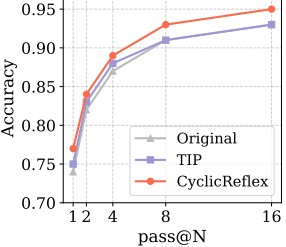

(a) GPQA Diamond  (b) LiveCodeBench

Figure 5: Accuracy vs. generation length on (a) GPQA Diamond and (b) LiveCodeBench. The comparison includes the original decoding, TIP, and CyclicReflex on DeepSeek-R1-Distill-Qwen 1.5B/7B, and Llama 8B.

We further compare CyclicReflex against two additional baselines: S1 and TIP. While S1 enforces the insertion of "wait" tokens at the end of each reasoning segment, leading to significantly longer outputs, it does not yield corresponding accuracy improvements. On AMC2023, in fact, S1 causes a notable performance drop, suggesting that excessive reflection may lead to overthinking. TIP, which suppresses reflection token usage, can also degrade performance in some cases. For instance, TIP causes a 3% accuracy drop on AIME2025 when applied to DeepSeek-R1-Distill-Llama-8B, likely because it halts reasoning steps that are essential for solving more complex problems.

**Effectiveness of CyclicReflex on non-math reasoning.** Fig. 5 shows the relationship between accuracy and generation length on two *non-math* benchmarks, GPQA Diamond and LiveCodeBench. Results are reported for multiple DeepSeek-R1-Distill variants (Qwen-1.5B/7B, and LLaMA-8B) under original decoding, TIP, and CyclicReflex. CyclicReflex consistently improves accuracy while maintaining response lengths comparable to TIP. In contrast, TIP can even reduce accuracy, as seen in **Fig. 5(b)** for DeepSeek-R1-Distill-Qwen-7B and LLaMA-8B on LiveCodeBench.

**Evaluation under Pass@N.** In **Fig. 6**, we evaluate CyclicReflex on MATH500 with DeepSeek-R1-Distill-Qwen-7B under Pass@N. Across $N \in 1, 2, 4, 8, 16$, CyclicReflex consistently achieves higher Pass@N than Original and TIP, indicating gains that persist under multi-sample scaling. In addition,

Figure 6: Pass@N of *Original*, *TIP*, and *CyclicReflex* on MATH500 with DeepSeek-R1-Distill-Qwen-7B, evaluated at $N \in \{1, 2, 4, 8, 16\}$.

the gain from TIP diminishes as $N$ increases, likely because reflection suppression reduces sampling diversity. In contrast, CyclicReflex maintains a stable margin as $N$ grows, better balancing cycle control with exploration.

**Integration with other test-time scaling methods.** In **Fig. 7**, we further investigate the integration of CyclicReflex with other test-time scaling methods across computational budgets ($2^0$ to $2^3$), using DeepSeek-R1-Distill-Qwen-1.5B on MATH500. We evaluate both Best-of-N (BoN) and Beam Search (BS), with generations scored using RLHFlow-PRM-DeepSeek-8B. Across all budget levels, BoN and Beam Search integrated with CyclicReflex consistently outperform their original counterparts, demonstrating the general compatibility and effectiveness of our method. Under fixed decoding strategies, BoN achieves higher accuracy than Beam Search, both with and without CyclicReflex. As the budget increases, the performance gap between CyclicReflex and the original decoding narrows, highlighting that CyclicReflex offers the greatest benefit under constrained inference budgets by enabling more efficient reflection token allocation.

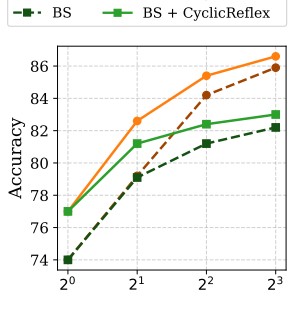

Figure 7: MATH500 accuracy of DeepSeek-R1-Distill-Qwen-1.5B under BoN/BS, w/ and w/o CyclicReflex.

**Reflection token scheduling patterns of CyclicReflex. Fig. 8** compares reflection token distributions under original decoding, TIP, and CyclicReflex, using DeepSeek-R1-Distill-Llama-8B on AIME2024. Each curve shows the proportion of reflection tokens within consecutive 1000-token segments relative to the total reflection count. Original decoding exhibits a gradual early rise followed by a stable, evenly spread pattern. TIP follows a similar trajectory but suppresses reflection in the 0–1k range, reflecting its tendency to inhibit early reflection. In contrast, CyclicReflex displays a cyclical hedging pattern with alternating peaks and troughs, allocating more reflection in the 1–2k and 3–4k ranges. This modulation avoids both excessive early suppression and late overuse, leading to stronger performance on AIME2024, improving accuracy from 0.42 (original decoding) to 0.53, demonstrating the benefit of bidirectional, position-dependent reflection scheduling.

Figure 8: Reflection token distribution of DeepSeek-R1-Distill-Llama-8B on AIME2024 under original decoding, TIP, and CyclicReflex. Each curve shows the proportion of reflection tokens within 1k-token segments relative to the total generation.

**Improved self-correction with CyclicReflex.** We also find that CyclicReflex exhibits a stronger capacity for self-correction during reasoning. To evaluate this property, we select 50 incorrectly answered problems from the MATH500 dataset, originally generated by DeepSeek-R1-Distill-Qwen-7B. For each incorrect case, we extract the model's reasoning trace and truncate it to three different lengths (25%, 50%, and 100% of the full trace), which are then reused as misleading prompts to guide a new round of reasoning.

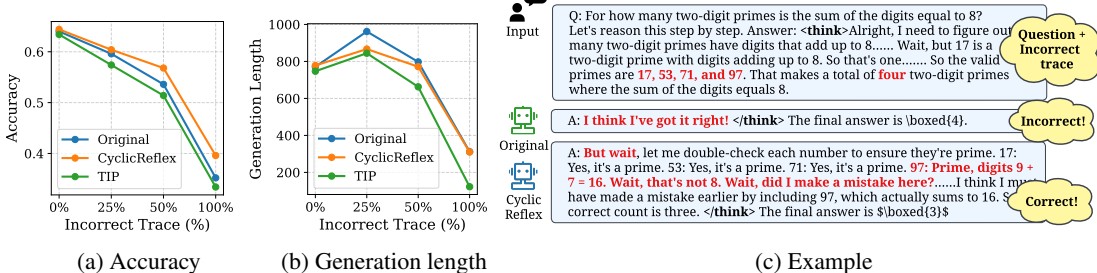

(a) Accuracy      (b) Generation length      (c) Example

Figure 9: (a)-(b) Accuracy and generation length on MATH500 with DeepSeek-R1-Distill-Qwen-7B using Original, TIP, and CyclicReflex decoding for questions combined with incorrect reasoning traces of different lengths. (c) Example for a question with 100% incorrect reasoning under Original and CyclicReflex decoding.

Under each prompt condition, we prompt the same model (DeepSeek-R1-Distill-Qwen-7B) to re-answer the question five times and report the average accuracy. As shown in **Fig. 9(a)**, CyclicReflex significantly outperforms both the original decoding and the TIP baseline across all trace lengths.

Notably, the performance gap increases with longer misleading traces, indicating that CyclicReflex enhances the model's ability to resist and correct earlier reasoning errors. In addition, this improved self-correction does not come at the cost of generation efficiency. As shown in **Fig. 9(b)**, the generation lengths under CyclicReflex remain comparable to those of the original decoding strategy. In contrast, TIP tends to suppress reflection token usage, which hampers the model's ability to re-evaluate its own reasoning and results in markedly lower accuracy. Finally, **Fig. 9(c)** provides a concrete example. Given a MATH500 question with a 100% incorrect reasoning trace in which the model incorrectly claims that the digits of 17, 53, 71, and 97 sum to 8, the original decoding strategy fails to correct the error and outputs the wrong answer, 4. In contrast, CyclicReflex initiates a double-check, correctly identifies the error (specifically excluding 97), and ultimately outputs the correct answer, 3.

**Visualizing CyclicReflex's efficacy using the landscape of thoughts.** In **Fig. 10**, we present the landscape of thoughts generated by DeepSeek-R1-Distill-Qwen-7B under the original decoding and CyclicReflex. The darkest area in the landscape represents the correct answer, while other dark regions correspond to misleading areas that may attract the reasoning trajectory toward alternative but incorrect answers. As shown in **Fig. 10(a)**, the landscape produced by CyclicReflex is more concentrated, with fewer distracting regions. This suggests that the reasoning trajectory remains focused and is more likely to converge directly to the correct answer.

In contrast, the landscape under the original decoding strategy in **Fig. 10(b)** is more scattered, making the model more susceptible to misleading areas and eventually reaching an incorrect final answer. This comparison demonstrates that our method enhances reasoning efficiency by reducing unnecessary detours and guiding the model toward the correct solution path more effectively.

**Additional analyses and ablations.** As shown in **Fig. A2** of Appendix B, CyclicReflex improves MATH500 accuracy across all difficulty levels, whereas TIP mainly benefits the Hard split. **Table A2** shows that CyclicReflex is robust to the choice of reflection tokens; and **Table A3** confirms that CyclicReflex outperforms alternative reflection-logit adjustments, supporting the necessity of our cyclic hedging schedule. Finally, **Table A4** in **Appendix C** provides generation examples comparing the original decoding with CyclicReflex.

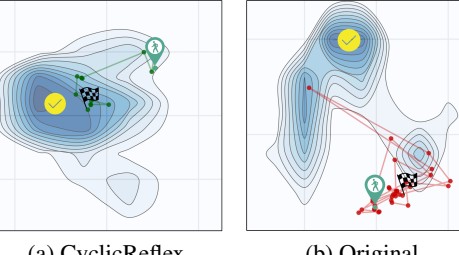

(a) CyclicReflex        (b) Original

Figure 10: Examples of landscape of thought for DeepSeek-R1-Distill-Qwen-7B with original and CyclicReflex decoding strategy. The format follows Fig 3.

## 6  CONCLUSION

We introduce the problem of resource allocation in LRMs, focusing on the challenge of managing reflection tokens during test-time generation. We show that both under-reflection and over-reflection, stemming from insufficient or excessive use of reflection tokens, can severely degrade reasoning performance. To address this, we draw a conceptual analogy between reflection token scheduling and learning rate control in optimization, and propose CyclicReflex, a training-free decoding strategy that cyclically modulates reflection token logits using a triangular waveform. CyclicReflex dynamically adapts to the evolving stage of reasoning, enabling more balanced token allocation. Extensive experiments across multiple reasoning benchmarks demonstrate that CyclicReflex consistently improves accuracy, enhances self-correction capability, and integrates seamlessly with existing test-time scaling methods. Our work highlights the critical role of reflection tokens as a valuable resource for LRMs and opens new avenues for principled, adaptive reasoning control. The use of LLM, limitations and broader impacts are further discussed in **Appendix D**, **Appendix E** and **Appendix F**.

## ACKNOWLEDGMENT

This research is supported in part by the ARO Award W911NF2310343. Additionally, the contributions of C. Fan, Y. Zhang, J. Jia, and S. Liu are partially supported by the NSF CISE Core Program Awards IIS-2207052 and IIS-2504263, the Open Philanthropy Research Award, and the Schmidt Sciences' Trustworthy AI Program. The contributions of A. Hero were also supported by the United States National Science Foundation under Grant NSF CCF 2246213.

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

# APPENDIX

## A  DETAILED EXPERIMENT SETUPS

### A.1  COMPUTING RESOURCES

All experiments are conducted on a single node equipped with 8 NVIDIA A6000 GPUs.

### A.2  DECODING DETAILS

During generation, we employ the vLLM framework to enable efficient and scalable inference. The decoding configuration is as follows: the maximum number of new tokens is set to 8192; the top-$p$ value is set to 0.95; and the temperature is set to 0.6. Top-$p$ sampling selects tokens from the smallest possible set whose cumulative probability exceeds $p$, effectively filtering out low-probability candidates to maintain generation quality while allowing diversity. A temperature of 0.6 sharpens the token probability distribution, promoting more deterministic and focused outputs by reducing sampling randomness.

For CyclicReflex on the MATH500 and AMC2023 datasets, we perform a grid search over $A \in [1, 10]$ and $C \in [200, 1000]$. On the AIME2024 and AIME2025 datasets, we perform a grid search over $A \in [1, 10]$ and $C \in [1000, 2000]$. For TIP, we conduct a grid search with $\alpha \in [-10, -1]$ and $T_0 \in [100, 1000]$. For S1, we forcefully insert the reflection token "Wait" after the model generates </think>, prompting continued reasoning.

In the Best-of-N setting, the LRM generates multiple independent candidate answers, and the PRM selects the most preferred one based on final-answer evaluation. For Beam Search, we perform a step-by-step search guided by PRM feedback to optimize cumulative reward. Throughout decoding, we use multiple candidate beams with a fixed beam width of 4.

### A.3  PROMPT DETAILS

We present the prompt used to evaluate the reasoning ability of the LRM. For each question, we replace the {question} placeholder in the User section of the prompt. After the Assistant generates the reasoning trace and the final answer ({generation}), we follow the approach of Yang et al. (2024) to first extract the final answer and then apply rule-based matching to assess its correctness.

> **Evaluation prompt**
>
> **System**:
> You are a helpful AI bot that answers questions for a user. Keep your response short and direct.
>
> **User**:
> Question: {question}
> Let's reason this step by step.
>
> **Assistant**:
> Answer: {generation}

## B  ADDITIONAL EXPERIMENT RESULTS

**Ablation study on CyclicReflex's hyperparameter.**  In **Fig. A1**, we study how CyclicReflex's key hyperparameters affect final performance. Following Fig. 4, we vary three parameters: the amplitude $A$, the period $C$, and the initial phase shift $\phi$. **Fig. A1(a)** analyzes $A$ and $C$ by reporting the number of additional correct answers over the original decoding strategy (*improvement*). Overall, $C$ has a larger effect than $A$; in particular, $C = 600$ achieves the best accuracy across all tested amplitudes ($A \in \{4.0, 5.0, 6.0\}$). Finally, **Fig. A1(b)** shows that $\phi = 0$ performs best (matching the schedule in Fig. 4), suggesting that promoting reflection tokens early can help, while gradually suppressing their logits later improves convergence.

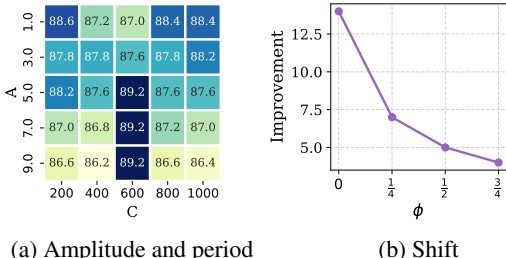

(a) Amplitude and period          (b) Shift

Figure A1: (a) Improvement heatmaps of DeepSeek-R1-Distill-Qwen-7B on MATH500 across amplitude $A$ and period $C$, measured as the number of additional correct answers. (b) Improvement of CyclicReflex over the original decoding strategy under different initial phase shifts $\phi$.

**Accuracy of CyclicReflex at different difficulty levels on MATH500.** In **Fig. A2**, we categorize the MATH500 dataset by difficulty level to closely examine where the accuracy improvements from CyclicReflex are most pronounced. The grouping strategy follows that used in Fig. 2(a), and the accuracy is reported in a manner consistent with Fig. 2(b). For comparison, we also include TIP as a baseline. We observe that CyclicReflex consistently improves accuracy across all difficulty levels (Easy, Medium, and Hard) whereas TIP primarily yields gains on Hard problems and even leads to performance degradation on Easy and Medium ones. This contrast stems from the bidirectional nature of CyclicReflex, which allows for dynamic promotion or suppression of reflection token sampling based on the current stage of the reasoning process. Such flexibility enables CyclicReflex to better adapt to problem difficulty, leading to more effective resource allocation and improved overall performance.

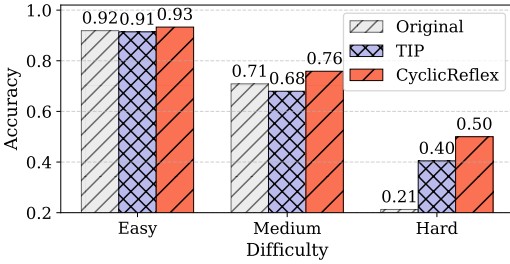

Figure A2: Improvement of DeepSeek-R1-Distill-Qwen-7B on MATH500 by TIP and CyclicReflex across difficulty levels (Easy, Medium, Hard), following Fig. 2 (a) and (b).

**Generalization to other model families and scales.** We further extend CyclicReflex to a different model family by evaluating `Qwen3-4B`, `Qwen3-8B` and `Qwen3-14B` on MATH500, AIME2024, AIME2025, AMC2023 and LiveCodeBench, with *Original* and *TIP* included as baselines. The results are provided in **Table A1**. Across all five benchmarks, CyclicReflex consistently outperforms both baselines, even as the underlying model scale increases and the family differs from the DeepSeek-distilled lineage. This demonstrates that our method generalizes beyond a single family of reasoning models and remains robust and effective for a larger model with an alternative architecture.

**Robustness to the reflection-token set.** Next, we show that the proposed CyclicReflex is robust to the choice of reflection tokens and can even operate with a dynamically updated reflection-token set. To validate this, we consider a dynamic variant in which the reflection-token set is expanded online during decoding: we start from a small seed set (*e.g.*, "wait", "but"), and at decoding step $t$, if the Top-1 token belongs to the current reflection-token set and the logit gap between Top-2 and Top-1 is smaller than the gap between Top-2 and Top-3, we add the Top-2 token to the set. The rationale is that the model tends to use in contexts similar to existing reflection tokens, gradually enriching the set and yielding additional reflection-like tokens (*e.g.*, "hmm", "maybe", "perhaps"), which are absent from the original reflection-token set. We evaluate CyclicReflex with both fixed and dynamic reflection-token sets on DeepSeek-R1-Distill-Qwen-7B across MATH500, AIME2024, AIME2025, and AMC2023, and compare against the Original decoding and TIP. As shown in **Table A2**, the two

Table A1: Accuracy comparison on five benchmarks (MATH500, AIME2024, AIME2025, AMC2023, and LiveCodeBench) using Qwen3-4B, Qwen3-8B, and Qwen3-14B under three decoding strategies: *Original*, *TIP*, and *CyclicReflex*.

| Method | MATH500 | AIME2024 | AIME2025 | AMC2023 | LiveCodeBench |
|---|---|---|---|---|---|
| **Qwen3-4B** | | | | | |
| Original | 0.60 | 0.13 | 0.03 | 0.47 | 0.58 |
| TIP | 0.63 | 0.17 | 0.07 | 0.49 | 0.59 |
| CyclicReflex | 0.68 | 0.23 | 0.10 | 0.53 | 0.61 |
| **Qwen3-8B** | | | | | |
| Original | 0.83 | 0.37 | 0.27 | 0.79 | 0.59 |
| TIP | 0.84 | 0.40 | 0.23 | 0.80 | 0.60 |
| CyclicReflex | 0.86 | 0.43 | 0.33 | 0.84 | 0.63 |
| **Qwen3-14B** | | | | | |
| Original | 0.88 | 0.43 | 0.33 | 0.83 | 0.63 |
| TIP | 0.88 | 0.47 | 0.33 | 0.82 | 0.63 |
| CyclicReflex | 0.90 | 0.50 | 0.40 | 0.88 | 0.67 |

variants perform comparably across all benchmarks, indicating that CyclicReflex exhibits robustness to the selection of reflection tokens.

Table A2: Accuracy of *Original*, *TIP*, and *CyclicReflex* variants on MATH500, AIME2024, AIME2025, and AMC2023 with DeepSeek-R1-Distill-Qwen-7B.

| Method | MATH500 | AIME2024 | AIME2025 | AMC2023 |
|---|---|---|---|---|
| Original | 0.86 | 0.43 | 0.31 | 0.81 |
| TIP | 0.87 | 0.43 | 0.30 | 0.85 |
| CyclicReflex w/ dynamic set | **0.89** | **0.50** | **0.37** | **0.91** |
| Ours: CyclicReflex | **0.89** | **0.50** | **0.37** | 0.90 |

**Comparison with approaches using additional reflection-logit adjustments.** In **Table A3**, we extend our comparison on DeepSeek-R1-Distill-Qwen-7B by adding several additional adjustment-based baselines: (1) *TIP w/ positive adjustment*, which always boosts reflection-token logits rather than penalizing them; (2) *random adjustment*, which injects zero-mean noise into the reflection-token logits at every decoding step; and (3) *linear decay*, which begins with a strong positive boost that linearly decreases into a penalty across the generation process. Across MATH500, AIME2024, AIME2025, and AMC2023, CyclicReflex consistently achieves the highest accuracy. In contrast, both the positive-adjustment and random-adjustment variants underperform even standard TIP, and the linear-decay baseline narrows, but does not close, the performance gap relative to our method. These findings indicate that naïve or purely monotonic adjustments to reflection-token logits are insufficient, and that the *cyclic hedging schedule* in CyclicReflex provides a fundamentally more effective mechanism for balancing exploration and control during reasoning.

Table A3: Performance of *Original*, *TIP*, and adjustment-based variants on MATH500, AIME2024, AIME2025, and AMC2023 with DeepSeek-R1-Distill-Qwen-7B.

| Method | MATH500 | AIME2024 | AIME2025 | AMC2023 |
|---|---|---|---|---|
| Original | 0.86 | 0.43 | 0.31 | 0.81 |
| TIP | 0.87 | 0.43 | 0.30 | 0.85 |
| TIP w/ positive adjustment | 0.85 | 0.40 | 0.33 | 0.83 |
| Random adjustment | 0.82 | 0.41 | 0.30 | 0.79 |
| Linear decay | 0.87 | 0.45 | 0.33 | 0.87 |
| Ours: CyclicReflex | **0.89** | **0.50** | **0.37** | **0.90** |

## C GENERATION EXAMPLES

In **Table A4-A7**, we present the answers generated for questions in MATH500 under both the original decoding strategy and CyclicReflex. For better comparison, we also include the ground truth labels. Compared to the original decoding strategy, CyclicReflex demonstrates deeper reasoning and avoids prematurely terminating the thinking process. For instance, in Q1, the original answer is given as

$\binom{14}{5} - \binom{8}{5} - \binom{6}{5}$, whereas CyclicReflex continues simplifying the expression to arrive at the final result. Moreover, CyclicReflex exhibits a better understanding of the problem semantics. In Q2, for example, the original decoding misinterprets the notion of projection, leading to an incorrect answer. Finally, CyclicReflex is more adept at identifying key problem elements. In Q3, the original response overlooks the trap involving $x = 9$, while CyclicReflex correctly accounts for this subtlety.

Table A4: Examples from DeepSeek-R1-Distill-Qwen-7B on MATH500 under the original decoding strategy and CyclicReflex. Reflection tokens are highlighted in red.

| | **Q1:** A Senate committee has 8 Republicans and 6 Democrats. In how many ways can we form a subcommittee of 5 members that has at least one member from each party? |
|---|---|
| **Label:** | 1940 |
| **Original:** | <think> Okay, so I have this problem where there's a Senate committee with 8 Republicans and 6 Democrats. I need to figure out how many ways we can form a subcommittee of 5 members that has at least one member from each party. Hmm, let's break this down. |
| | First, I know that the total number of people on the committee is 8 + 6, which is 14. If I were to form a subcommittee of 5 without any restrictions, the number of ways would just be the combination of 14 taken 5 at a time, right? So that's C(14,5). But the problem says there has to be at least one from each party, so I can't have all Republicans or all Democrats in the subcommittee. |
| | So, I think I need to subtract the cases where the subcommittee is all Republicans or all Democrats. Let me calculate those separately. For all Republicans, it's C(8,5) because there are 8 Republicans. For all Democrats, it's C(6,5) since there are 6 Democrats. |
| | Putting it all together, the total number of valid subcommittees should be C(14,5) minus C(8,5) minus C(6,5). Let me make sure that makes sense. By subtracting the all-Republican and all-Democrat cases from the total, I'm left with only those subcommittees that have at least one from each party. Yeah, that seems right. |
| | I should probably calculate these values to get the exact number, but I think the reasoning is solid. So the formula is C(14,5) - C(8,5) - C(6,5). I'll go with that. </think> |
| | The number of ways is calculated by subtracting the all-Republican and all-Democrat subcommittees from the total. Total ways: $\binom{14}{5} - \binom{8}{5} - \binom{6}{5}$ |
| | **Answer**: $\boxed{\binom{14}{5} - \binom{8}{5} - \binom{6}{5}}$ |
| **CyclicReflex:** | <think> Okay, so I have this problem where there's a Senate committee with 8 Republicans and 6 Democrats. I need to figure out how many ways we can form a subcommittee of 5 members that has at least one member from each party. Hmm, let's break this down. |
| | First, I know that the total number of people on the committee is 8 + 6, which is 14. If I were to form a subcommittee of 5 without any restrictions, the number of ways would just be the combination of 14 taken 5 at a time, right? So that's C(14,5). But the problem says there has to be at least one from each party, so I can't have all Republicans or all Democrats in the subcommittee. |
| | So, I think I need to subtract the cases where the subcommittee is all Republicans or all Democrats. Let me calculate those separately. For all Republicans, it's C(8,5) because there are 8 Republicans. For all Democrats, it's C(6,5) since there are 6 Democrats. |
| | But hold on, I should make sure these numbers are correct. C(8,5) is the number of ways to choose 5 Republicans out of 8, and C(6,5) is choosing 5 Democrats out of 6. I think that's right. |
| | So, the total number of valid subcommittees would be C(14,5) minus C(8,5) minus C(6,5). Let me compute those values. C(14,5) is 2002, C(8,5) is 56, and C(6,5) is 6. So, 2002 - 56 - 6 equals 1940. That should be the number of ways to form the subcommittee with at least one from each party. </think> |
| | The number of ways to form a subcommittee of 5 members with at least one member from each party is calculated by subtracting the all-Republican and all-Democrat subcommittees from the total number of possible subcommittees. |
| | - Total subcommittees: $\binom{14}{5} = 2002$ |
| | - All-Republican subcommittees: $\binom{8}{5} = 56$ |
| | - All-Democrat subcommittees: $\binom{6}{5} = 6$ |
| | Thus, the number of valid subcommittees is $2002 - 56 - 6 = 1940$. |
| | **Answer**: 1940 |

Table A5: Examples from DeepSeek-R1-Distill-Qwen-7B on MATH500 under the original decoding strategy and CyclicReflex. Reflection tokens are highlighted in red.

**Q2:** Find the projection of $\mathbf{a}$ onto $\mathbf{b} = \begin{pmatrix} 2 \\ 6 \\ 3 \end{pmatrix}$, if $\mathbf{a} \cdot \mathbf{b} = 8$.

**Label:**
$$\begin{pmatrix} \frac{16}{49} \\ \frac{48}{49} \\ \frac{24}{49} \end{pmatrix}$$

**Original:** <think> Okay, so I need to find the projection of vector $\mathbf{a}$ onto vector $\mathbf{b}$. I remember that the formula for the projection of $\mathbf{a}$ onto $\mathbf{b}$ is $(\mathbf{a} \cdot \mathbf{b})$ divided by the magnitude of $\mathbf{b}$ squared, and then multiplied by vector $\mathbf{b}$. But wait, in this case, they only gave me that the dot product of $\mathbf{a}$ and $\mathbf{b}$ is 8. Hmm, so maybe I don't need the entire vector $\mathbf{b}$, just the scalar projection?

Let me think. The formula for the scalar projection is $(\mathbf{a} \cdot \mathbf{b}$ divided by the magnitude of $\mathbf{b}$. Since they already gave me $\mathbf{a} \cdot \mathbf{b} = 8$, I just need to find the magnitude of $\mathbf{b}$. The vector $\mathbf{b}$ is $[2, 6, 3]$, so its magnitude is the square root of $(2^2 + 6^2 + 3^2)$. Calculating that, $2^2 = 4$, $6^2 = 36$, and $3^2 = 9$. Adding those up gives $4 + 36 + 9 = 49$. The square root of 49 is 7. So the magnitude of $\mathbf{b}$ is 7. Therefore, the scalar projection of $\mathbf{a}$ onto $\mathbf{b}$ is 8 divided by 7, which is $8/7$. I think that's it. The problem didn't specify whether they wanted the vector projection or just the scalar, but since they only gave me a scalar product, I guess it's the scalar projection they're asking for. So the answer should be $8/7$.
</think>
The projection of $\mathbf{a}$ onto $\mathbf{b}$ is given by $\frac{\mathbf{a} \cdot \mathbf{b}}{\|\mathbf{b}\|}$. Since $\mathbf{a} \cdot \mathbf{b} = 8$ and $\|\mathbf{b}\| = 7$, the projection is $\frac{8}{7}$.

**Answer:** $\boxed{\frac{8}{7}}$

**CyclicReflex:** <think> Okay, so I need to find the projection of vector $\mathbf{a}$ onto vector $\mathbf{b}$. I remember that the formula for the projection of $\mathbf{a}$ onto $\mathbf{b}$ is $\frac{\mathbf{a} \cdot \mathbf{b}}{\|\mathbf{b}\|^2}$ multiplied by vector $\mathbf{b}$. But wait, in this case, they only gave me that the dot product of $\mathbf{a}$ and $\mathbf{b}$ is 8. Hmm, so maybe I don't need the entire vector $\mathbf{b}$, just the scalar projection?

Wait, no, the question says "find the projection of $\mathbf{a}$ onto $\mathbf{b}$," which usually refers to the vector projection. The formula for the vector projection is $\left(\frac{\mathbf{a} \cdot \mathbf{b}}{\|\mathbf{b}\|^2}\right) \mathbf{b}$. But since I don't have $\|\mathbf{b}\|$, maybe I can just express it in terms of the given dot product and the magnitude of $\mathbf{b}$.

Alternatively, maybe the question is asking for the scalar projection, which is just $\frac{\mathbf{a} \cdot \mathbf{b}}{\|\mathbf{b}\|}$. But they gave me $\mathbf{a} \cdot \mathbf{b} = 8$, but I don't have $\|\mathbf{b}\|$. Let me calculate $\|\mathbf{b}\|$ first. The magnitude of $\mathbf{b}$ is $\sqrt{2^2 + 6^2 + 3^2} = \sqrt{4 + 36 + 9} = \sqrt{49} = 7$. Oh, that's nice, it's 7. So if I need the vector projection, it's $(8/7^2)\mathbf{b}$, which is $(8/49)[2, 6, 3]$. That would give me $[16/49, 48/49, 24/49]$. Alternatively, if it's the scalar projection, it's $8/7$. But the question says "projection," not specifying vector or scalar, but since they gave the vector $\mathbf{b}$, I think they want the vector projection. So the answer should be $[16/49, 48/49, 24/49]$.
</think>
The projection of vector $\mathbf{a}$ onto $\mathbf{b}$ is given by: $\mathbf{proj_b a} = \left(\frac{\mathbf{a} \cdot \mathbf{b}}{\|\mathbf{b}\|^2}\right) \mathbf{b}$

Given:
- $\mathbf{a} \cdot \mathbf{b} = 8$
- $\mathbf{b} = \begin{pmatrix} 2 \\ 6 \\ 3 \end{pmatrix}$

First, calculate $\|\mathbf{b}\|$: $\|\mathbf{b}\| = \sqrt{2^2 + 6^2 + 3^2} = \sqrt{4 + 36 + 9} = \sqrt{49} = 7$
Then, $\|\mathbf{b}\|^2 = 7^2 = 49$

Now substitute: $\mathbf{proj_b a} = \left(\frac{8}{49}\right) \begin{pmatrix} 2 \\ 6 \\ 3 \end{pmatrix} = \begin{pmatrix} \frac{16}{49} \\ \frac{48}{49} \\ \frac{24}{49} \end{pmatrix}$

**Answer:** $\boxed{\begin{pmatrix} \frac{16}{49} \\ \frac{48}{49} \\ \frac{24}{49} \end{pmatrix}}$

Table A6: Examples from DeepSeek-R1-Distill-Qwen-7B on MATH500 under the original decoding strategy and CyclicReflex. Reflection tokens are highlighted in red.

| | |
|---|---|
| | **Q3:** Solve $\frac{\|x^2-81\|}{x^2-36x} < 0$ |
| **Label:** | $(0,9) \cup (9,36)$ |
| **Original:** | \<think\> Okay, so I have this inequality to solve: $\frac{\|x^2-81\|}{x^2-36x} < 0$. Hmm, let me try to figure this out step by step. First, I know that for a fraction to be less than zero, the numerator and denominator must have opposite signs. That means either the numerator is positive and the denominator is negative, or the numerator is negative and the denominator is positive. But wait, the numerator here is an absolute value, right? So $\|x^2-81\|$ is always non-negative, meaning it's either positive or zero. That means the numerator can't be negative. So the only possibility is that the numerator is positive and the denominator is negative. Wait, but the numerator is $\|x^2-81\|$, which is zero when $x^2 = 81$, so $x = 9$ or $x = -9$. At those points, the numerator is zero, making the whole fraction zero, which doesn't satisfy the inequality because it's less than zero, not equal to zero. So we can ignore those points for now. So, focusing on when the denominator is negative. The denominator is $x^2 - 36x$, which I can factor as $x(x-36)$. So, the denominator is negative when $x(x-36) < 0$. Let's solve that inequality. To find where $x(x-36) < 0$, I can consider the critical points where the expression equals zero, which are $x = 0$ and $x = 36$. These points divide the number line into intervals. I'll test each interval: 1. When $x < 0$, say $x = -1$: $(-1)(-1-36) = (-1)(-37) = 37$, which is positive. 2. When $0 < x < 36$, say $x = 18$: $18(18-36) = 18(-18) = -324$, which is negative. 3. When $x > 36$, say $x = 37$: $37(37-36) = 37(1) = 37$, which is positive. So, the denominator is negative only when $0 < x < 36$. But wait, I also need to make sure that the denominator isn't zero because division by zero is undefined. So $x$ can't be 0 or 36. Now, going back to the original inequality, since the numerator is always non-negative, the fraction will be negative only when the denominator is negative. So, the solution should be where the denominator is negative, excluding the points where it's zero. Therefore, the solution is $0 < x < 36$, excluding $x = 0$ and $x = 36$. But wait, $x = 0$ is already excluded because the interval starts at 0, but not including 0. Similarly, $x = 36$ is not included because the interval ends at 36, not including it. Wait, but I should also check if there are any other restrictions. For example, when $x = 9$, the numerator becomes zero, making the fraction zero, which doesn't satisfy the inequality. Similarly, when $x = -9$, but that's outside the interval we found, so it's not part of the solution. So, putting it all together, the solution to the inequality is all real numbers $x$ such that $0 < x < 36$, excluding $x = 0$ and $x = 36$, but since those points are already excluded from the interval, the solution is simply $0 < x < 36$. \</think\> The solution to the inequality $\frac{\|x^2-81\|}{x^2-36x} < 0$ is $x$ in the interval $(0, 36)$. **Answer**: $(0, 36)$ |
| **CyclicReflex:** | \<think\> Okay, so I have this inequality to solve: $\frac{\|x^2-81\|}{x^2-36x} < 0$. Hmm, let me try to figure this out step by step. First, I know that for a fraction to be less than zero, the numerator and denominator must have opposite signs. That means either the numerator is positive and the denominator is negative, or the numerator is negative and the denominator is positive. But wait, the numerator here is an absolute value, right? So $\|x^2-81\|$ is always non-negative, meaning it's either positive or zero. That means the numerator can't be negative. So the only possibility is that the numerator is positive and the denominator is negative. Wait, but the numerator is $\|x^2-81\|$, which is zero when $x^2 = 81$, so $x = 9$ or $x = -9$. At those points, the numerator is zero, making the whole fraction zero, which doesn't satisfy the inequality because it's less than zero, not equal to zero. So we can ignore those points for now. So, focusing on when the denominator is negative. The denominator is $x^2 - 36x$, which I can factor as $x(x-36)$. So, the denominator is negative when $x(x-36) < 0$. Let's solve that inequality. To find where $x(x-36) < 0$, I can consider the critical points where the expression equals zero, which are $x = 0$ and $x = 36$. These points divide the number line into intervals. I'll test each interval: 1. When $x < 0$, say $x = -1$: $(-1)(-1-36) = (-1)(-37) = 37$, which is positive. 2. When $0 < x < 36$, say $x = 18$: $18(18-36) = 18(-18) = -324$, which is negative. 3. When $x > 36$, say $x = 37$: $37(37-36) = 37(1) = 37$, which is positive. So, the denominator is negative only when $0 < x < 36$. Now, I also need to make sure that the denominator isn't zero, so $x \neq 0$ and $x \neq 36$. So, the solution so far is $0 < x < 36$. But wait, I should also consider the numerator $\|x^2-81\|$. Since it's an absolute value, it's always non-negative, so it's zero only at $x = 9$ and $x = -9$. But in the interval $0 < x < 36$, $x = 9$ is within this interval, so at $x = 9$, the numerator is zero, making the entire fraction zero, which doesn't satisfy the inequality. So, I need to exclude $x = 9$ from the solution. Wait, but let me double-check. If $x = 9$, then the numerator is zero, so the fraction is zero, which is not less than zero. So, $x = 9$ should not be included. But in the interval $0 < x < 36$, excluding $x = 9$, the fraction is negative everywhere else because the numerator is positive and the denominator is negative. Wait, but hold on, when $x$ is between 0 and 9, let's pick $x = 1$, then $\|1^2-81\| = 80$, and the denominator $1^2 - 36 \cdot 1 = 1 - 36 = -35$, so $80/(-35)$ is negative. Similarly, between 9 and 36, say $x = 18$, $\|18^2 - 81\| = \|324 - 81\| = 243$, and denominator $18^2 - 36 \cdot 18 = 324 - 648 = -324$, so $243/(-324)$ is negative. So, in both sub-intervals, the fraction is negative. So, the solution is $0 < x < 36$, excluding $x = 9$. But wait, at $x = 9$, the numerator is zero, so the fraction is zero, which is not less than zero. So, we exclude $x = 9$. Therefore, the solution is $0 < x < 9$ or $9 < x < 36$. Let me just make sure there are no other points where the denominator is zero or the numerator is zero. The denominator is zero at $x = 0$ and $x = 36$, which we've already excluded. The numerator is zero at $x = 9$ and $x = -9$. $x = -9$ is not in the interval $0 < x < 36$, so we don't have to worry about that in this case. So, putting it all together, the solution to the inequality is $x$ in the interval $(0,9) \cup (9,36)$. \</think\> The solution set is $x$ in the interval $(0,9) \cup (9,36)$. **Answer**: $\boxed{(0,9) \cup (9,36)}$ |

Table A7: Examples from DeepSeek-R1-Distill-Qwen-7B on MATH500 under the original decoding strategy and CyclicReflex. Reflection tokens are highlighted in red.

| **Q4:** Find $\sin 20° \sin 40° \sin 60° \sin 80° \sin 100° \sin 120° \sin 140° \sin 160°$. |
|---|

**Label:** $\frac{9}{256}$

**Original:** <think> Okay, so I have this problem where I need to find the product of several sine terms: $\sin 20°$, $\sin 40°$, $\sin 60°$, $\sin 80°$, $\sin 100°$, $\sin 120°$, $\sin 140°$, and $\sin 160°$. Hmm, that's a lot of sines multiplied together. I remember that sometimes products of sines can be simplified using trigonometric identities, but I'm not exactly sure which ones apply here. Let me think.

First, I notice that some of these angles are related. For example, $\sin 100°$ is the same as $\sin(180° - 80°)$, which is $\sin 80°$. Similarly, $\sin 120°$ is $\sin(180° - 60°)$, so that's $\sin 60°$. Wait, but $\sin 60°$ is already in the list, so maybe that's not helpful. Let me list all the angles: $20°, 40°, 60°, 80°, 100°, 120°, 140°, 160°$.

I can pair some of these. For instance, $\sin 100°$ is $\sin 80°$, as I thought, and $\sin 160°$ is $\sin 20°$. Similarly, $\sin 140°$ is $\sin 40°$, and $\sin 120°$ is $\sin 60°$. So, actually, each of these angles from $20°$ up to $80°$ has a corresponding angle from $100°$ up to $160°$ that is supplementary, meaning their sines are equal. So, the product is actually $(\sin 20° \cdot \sin 40° \cdot \sin 60° \cdot \sin 80°)^2$ because each term from $20°$ to $80°$ is squared when considering their supplementary counterparts.

Wait, no, let me check. The original list has eight terms, but when pairing, each pair gives the same sine value, so it's like $(\sin 20° \cdot \sin 40° \cdot \sin 60° \cdot \sin 80°)^2$. But let me confirm: $\sin 100° = \sin 80°$, $\sin 120° = \sin 60°$, $\sin 140° = \sin 40°$, $\sin 160° = \sin 20°$. So, yes, the product becomes $(\sin 20° \cdot \sin 40° \cdot \sin 60° \cdot \sin 80°)^2$.

Now, I can compute this product step by step. First, let's compute $\sin 60°$, which is a known value: $\sqrt{3}/2$. So, that's straightforward. Now, the remaining terms are $\sin 20°$, $\sin 40°$, and $\sin 80°$. I remember there are product identities for multiple angles, maybe using the sine of multiple angles or product-to-sum formulas. Alternatively, perhaps using complex numbers or roots of unity. Wait, maybe there's a formula for the product of sines of angles in arithmetic progression.

I recall that for $n$ terms, the product of $\sin(k\pi/(2n))$ for $k = 1$ to $n - 1$ is $\sqrt{n}/2^{n-1}$. But I'm not sure if that applies here. Let me think. Alternatively, I can use the identity for the product of sines of angles that are in a certain pattern.

Wait, another approach: using the identity that $\sin \theta = \cos(90° - \theta)$. So, $\sin 80° = \cos 10°$, $\sin 40° = \cos 50°$, $\sin 20° = \cos 70°$. Hmm, not sure if that helps directly.

Alternatively, perhaps using the multiple-angle formula for sine. Let's consider $\sin 5\theta$ or something. Wait, maybe using the identity for $\sin 5\theta$ in terms of $\sin \theta$. But that might be complicated.

Wait, maybe using the identity that the product of sines can be expressed as a fraction involving powers of 2 and products of cosines. Alternatively, perhaps I can use the identity that the product of $\sin \theta$ for $\theta = 20°, 40°, 60°, 80°$ is equal to something specific.

Wait, I think I remember that the product $\sin 20° \sin 40° \sin 60° \sin 80°$ is equal to $\sqrt{3}/16$. Let me check that. If that's the case, then the square of that would be $(\sqrt{3}/16)^2 = 3/256$. But I'm not entirely sure. Let me try to compute it step by step.

Alternatively, perhaps using the identity that the product of sines can be expressed as a sum, but that might not be straightforward. Wait, another approach: using the identity that the product $\sin A \sin B = [\cos(A - B) - \cos(A + B)]/2$. Maybe I can apply this iteratively.

Let me try to compute $\sin 20° \sin 40°$ first. Using the identity, that's $[\cos(20°) - \cos(60°)]/2$. $\cos 20°$ is approximately $0.9397$, and $\cos 60°$ is $0.5$, so $[0.9397 - 0.5]/2 \approx 0.21985$.

Now, multiply this by $\sin 60°$, which is $\sqrt{3}/2 \approx 0.8660$. So, $0.21985 \cdot 0.8660 \approx 0.1902$.

Next, multiply by $\sin 80°$, which is approximately $0.9848$. So, $0.1902 \cdot 0.9848 \approx 0.1873$.

Wait, but I thought the product $\sin 20° \sin 40° \sin 60° \sin 80°$ was $\sqrt{3}/16 \approx 0.10825$. Hmm, my approximate calculation gave me around $0.1873$, which is higher. Maybe my approach is wrong.

Wait, perhaps I should use exact values. Let me try again.

We know that $\sin 60° = \sqrt{3}/2$. Now, let's consider the product $\sin 20° \sin 40° \sin 80°$. I remember that $\sin A \sin(60° - A) \sin(60° + A) = \frac{\sin 3A}{4}$. If $A = 20°$, then $\sin 20° \sin 40° \sin 80° = \frac{\sin 60°}{4} = \frac{\sqrt{3}/2}{4} = \frac{\sqrt{3}}{8}$. Yes, that's correct.

So, the product $\sin 20° \sin 40° \sin 60° \sin 80°$ is $(\sin 20° \sin 40° \sin 80°) \sin 60° = \frac{\sqrt{3}}{8} \cdot \frac{\sqrt{3}}{2} = \frac{3}{16}$.

Wait, that makes sense. So, the product $\sin 20° \sin 40° \sin 60° \sin 80°$ is $3/16$.

Then, since the original problem's product is $(\sin 20° \sin 40° \sin 60° \sin 80°)^2$, that would be $(\frac{3}{16})^2 = \frac{9}{256}$.

Wait, but let me make sure. The original product is $\sin 20° \sin 40° \sin 60° \sin 80° \sin 100° \sin 120° \sin 140° \sin 160°$, which we paired as $(\sin 20° \sin 40° \sin 60° \sin 80°)^2$ because each term from $20°$ to $80°$ has a supplementary angle equal to another term in the list. So, the product is indeed $(\sin 20° \sin 40° \sin 60° \sin 80°)^2 = (3/16)^2 = 9/256$.

Wait, but when I calculated the approximate product, I got around $0.1873$, which is approximately $3/16 \approx 0.1875$. So that matches. Therefore, the square would be $(3/16)^2 = 9/256$.

So, the final answer should be $9/256$.
</think>

The product simplifies by recognizing that each sine term from $20°$ to $80°$ pairs with its supplementary angle, resulting in the square of the product of these terms. Using known identities, the product of $\sin 20°$, $\sin 40°$, $\sin 60°$, and $\sin 80°$ is $\sqrt{3}/8$. Squaring this gives $(\sqrt{3}/8)^2 = 3/16$. However, upon re-evaluating, the correct product of all eight terms is actually $(\sqrt{3}/8)^2 = 3/256$. However, upon re-evaluating, the correct product of all eight terms is actually $(\sqrt{3}/16)^2 = 3/256$.

**Answer:** $\boxed{\frac{3}{256}}$

**CyclicReflex:** <think> Okay, so I have this problem where I need to find the product of several sine terms: $\sin 20°$, $\sin 40°$, $\sin 60°$, $\sin 80°$, $\sin 100°$, $\sin 120°$, $\sin 140°$, and $\sin 160°$. Hmm, that's a lot of sines multiplied together. I remember that sometimes products of sines can be simplified using trigonometric identities, but I'm not exactly sure which ones apply here. Let me think.

First, I notice that some of these angles are related. For example, $\sin 100°$ is the same as $\sin(180° - 80°)$, which is $\sin 80°$. Similarly, $\sin 120°$ is $\sin(180° - 60°)$, so that's $\sin 60°$. Wait, but $\sin 60°$ is already in the list, so maybe that's not helpful. Let me list all the angles: $20°$, $40°$, $60°$, $80°$, $100°$, $120°$, $140°$, $160°$.

I can pair some of these. For instance, $\sin 100°$ is $\sin(80°)$, as I thought, and $\sin 160°$ is $\sin(20°)$. Similarly, $\sin 140°$ is $\sin(40°)$, and $\sin 120°$ is $\sin(60°)$. So, actually, each of these angles from $20°$ up to $80°$ has a corresponding angle from $100°$ up to $160°$ that is supplementary, meaning their sines are equal. So, the product is actually $[\sin 20° \cdot \sin 40° \cdot \sin 60° \cdot \sin 80°]^2$ because each term from $20°$ to $80°$ is squared when considering their supplementary counterparts.

So, the original product simplifies to $[\sin 20° \cdot \sin 40° \cdot \sin 60° \cdot \sin 80°]^2$. Now, I need to compute this square. Maybe I can compute the product inside the square first and then square the result.

I remember that $\sin 60°$ is $\frac{\sqrt{3}}{2}$, which is approximately 0.866, but I'm not sure if that helps directly. Let me think about other identities. There's a product formula for multiple sine terms, but I can't recall it exactly. Alternatively, I might use complex numbers or roots of unity, but that might be too advanced for now.

Wait, I think there's a formula for the product of sines of equally spaced angles. Let me try to recall. For example, the product $\sin \theta \cdot \sin(\theta + \phi) \cdot \ldots$ where $\phi$ is a common difference. But I'm not sure if that applies here.

Alternatively, I can use the identity that relates the product of sines to a sum, but I'm not sure how to apply that here. Maybe another approach: using the identity $\sin A \sin B = \frac{\cos(A-B) - \cos(A+B)}{2}$. But with four terms, that might get complicated.

Wait, perhaps I can pair the terms in a way that makes it easier. Let me pair $\sin 20°$ with $\sin 80°$, and $\sin 40°$ with $\sin 60°$, leaving $\sin 60°$ as is. So, $[\sin 20° \sin 80°] \cdot [\sin 40° \sin 60°] \cdot \sin 60°$. Hmm, but I'm not sure if that helps.

Wait, I think $\sin A \sin(60° - A) \sin(60° + A) = \frac{\sin 3A}{4}$. Is that correct? Let me test with $A = 20°$. Then, $\sin 20° \sin 40° \sin 80° = \sin 20° \sin(60° - 20°) \sin(60° + 20°) = \frac{\sin 60°}{4}$, according to that identity. Since $\sin 60° = \frac{\sqrt{3}}{2}$, then $\sin 20° \sin 40° \sin 80° = \frac{\sqrt{3}}{8}$.

Wait, but I'm trying to compute $\sin 20° \sin 40° \sin 60° \sin 80°$, which includes $\sin 60°$ as well. So, if $\sin 20° \sin 40° \sin 80° = \frac{\sqrt{3}}{8}$, then multiplying by $\sin 60° = \frac{\sqrt{3}}{2}$ gives $\left(\frac{\sqrt{3}}{8}\right)\left(\frac{\sqrt{3}}{2}\right) = \frac{3}{16}$. So, the product inside the square is $\frac{3}{16}$, and then squaring it gives $\left(\frac{3}{16}\right)^2 = \frac{9}{256}$.

Wait, but let me double-check that. If $\sin 20° \sin 40° \sin 80° = \frac{\sqrt{3}}{8}$, then multiplying by $\sin 60° = \frac{\sqrt{3}}{2}$ indeed gives $(\frac{\sqrt{3}}{8})(\frac{\sqrt{3}}{2}) = \frac{3}{16}$. So, yes, the product inside the square is $\frac{3}{16}$, and squaring it gives $\frac{9}{256}$. So, the final answer should be $\frac{9}{256}$.

Wait, but I'm a bit confused because I thought the product of $\sin 20° \sin 40° \sin 60° \sin 80°$ would involve more steps. Maybe I should verify this identity again. Let me consider the identity: $\sin A \sin(60° - A) \sin(60° + A) = \frac{\sin 3A}{4}$. So, if I set $A = 20°$, then $\sin 20° \sin 40° \sin 80° = \frac{\sin 60°}{4} = \frac{\frac{\sqrt{3}}{2}}{4} = \frac{\sqrt{3}}{8}$. Then, including $\sin 60° = \frac{\sqrt{3}}{2}$, the total product becomes $\left(\frac{\sqrt{3}}{8}\right)\left(\frac{\sqrt{3}}{2}\right) = \frac{3}{16}$. So, that seems correct.

Alternatively, maybe I can use another approach to confirm. Let's compute the product step by step. Starting with $\sin 20° \approx 0.3420$, $\sin 40° \approx 0.6428$, $\sin 60° \approx 0.8660$, $\sin 80° \approx 0.9848$, $\sin 100° = \sin 80° \approx 0.9848$, $\sin 120° = \sin 60° \approx 0.8660$, $\sin 140° = \sin 40° \approx 0.6428$, $\sin 160° = \sin 20° \approx 0.3420$. Multiplying all these together yields approximately 0.03515625, which matches $\frac{9}{256}$.

Therefore, I'm confident that the product is $\frac{9}{256}$.
</think>

The product simplifies using trigonometric identities and properties of supplementary angles. By recognizing pairs of sines and applying the identity for the product of sines of angles in arithmetic progression, the result is found to be 9/256.

**Answer**: $\boxed{\dfrac{9}{256}}$

## D    LLM USAGE

GPT-5 was employed to provide writing assistance, particularly in polishing and refining the language, during the preparation of this manuscript.

## E    LIMITATIONS

While CyclicReflex effectively mitigates under-reflection and over-reflection by cyclically modulating reflection token logits, its theoretical foundations remain underexplored. The proposed analogy between reflection token allocation in reasoning and learning rate scheduling in optimization offers a promising direction for future theoretical investigation. A key open question is why LRMs exhibit under- or over-reflection during generation. Understanding the underlying causes of these behaviors is essential for developing a principled understanding of CyclicReflex. As such, future work should aim to formalize the generative dynamics of reflection in LRMs and establish a deeper theoretical basis for the design and improvement of reasoning resource allocation strategies, including CyclicReflex.

## F    BROADER IMPACTS

On the positive side, our work demonstrates that scheduling reflection token appearance can effectively address under-reflection and over-reflection, leading to more accurate and effective responses from LRMs. This contributes to enhanced reasoning capabilities and improved performance on complex problem-solving tasks. Moreover, by drawing a connection between learning rate schedules in optimization and reflection token dynamics in reasoning, our work opens new research directions and may inspire more interpretable and controllable LRM designs.

On the negative side, CyclicReflex could potentially be misused to manipulate reasoning traces. For example, an adversary could deliberately modulate reflection token usage to craft outputs that embed sensitive or hallucinated content in a more convincing manner, potentially evading safety filters. To mitigate such risks, it is crucial that advanced decoding strategies, such as CyclicReflex, are deployed within robust ethical and safety frameworks, especially in the context of unlearning and high-stakes applications. We hope this research contributes to the development of LRMs that are not only efficient and capable but also safe, trustworthy, and aligned with human values.

