# OpenReview forum: "CyclicReflex: Improving Reasoning Models via Cyclical Reflection Token Scheduling"
_ICLR.cc/2026/Conference — ICLR 2026 Poster_

### Official Review · Reviewer_Fqv6 · 2025-10-14

**Soundness:** 2
**Presentation:** 3
**Contribution:** 2
**Rating:** 2
**Confidence:** 4

**Summary:**

The paper propose CycliReflex, a novel algorithm that control the generation of "reflection tokens" by adding a periodically changing penalty. Extensive experiments on math and other reasoning benchmarks are provided to show that the proposed algorithm is performing better than baselines like S1 and TIP.

**Strengths:**

- The paper is well-written.
- The algorithm is easy-to-follow and well-motivated.

**Weaknesses:**

- Only one class of models, i.e., Deepseek-distilled models are tested. Given the behavior of different model lineages could be substantially different, additional models, potentially larger ones, need to be covered.
- Certain highly related works are missing [1, 2, 3]. Specifically, [1, 2] are highly correlated papers that provide insights and potential baselines, while [3] is one highly similar paper that needs to be compared with.
- Some implementation details that are necessary for reproducibility are missing. Please refer to some of the questions for more details.
- For test-time scaling, no pass@N or cons@N results are provided, which are the more commonly used test-time scaling metrics.

[1]. Sadhukhan R, Chen Z, Zheng H, et al. Kinetics: Rethinking Test-Time Scaling Laws[J]. arXiv preprint arXiv:2506.05333, 2025.

[2]. Kang Z, Zhao X, Song D. Scalable best-of-n selection for large language models via self-certainty[J]. arXiv preprint arXiv:2502.18581, 2025.

[3]. Chen W, Koenig S, Dilkina B. Iterative Deepening Sampling as Efficient Test-Time Scaling[J]. arXiv preprint arXiv:2502.05449, 2025.

**Questions:**

1. How is LiveCodeBench tested? Specifically in code, the "reflection tokens" should not be included during the code generation process in the middle. How are such linguistic restrictions considered and affect the performance?
2. According to Appendix A, a grid search is needed to find the optimal hyperparameters used in the setting. Is the search done per-model per dataset or only once? And is there any overall recommendations for parameters when trying on new settings?
3. For accuracy results like Table 4, is the experiments averaged through multiple runs? Is it still 5 times or something else?

---

> ### Author Response · Authors · 2025-11-20
> **1. Response to additional model lineages and larger models (Weakness 1)**
>
> Thank you for pointing out the limitation of only evaluating DeepSeek-distilled models. We have added results on a different model family and a larger model: Qwen3 14B evaluated on MATH500, AIME2024, AIME2025, and AMC2023. We also report Original and TIP as baselines. The results are summarized in **Table R6**.
>
> Table R6. Accuracy of Qwen3 14B on MATH500, AIME2024, AIME2025, and AMC2023.
>
> |   Method    | MATH500 | AIME2024 | AIME2025 | AMC2023 |
> |-----------|:-------:|:--------:|:--------:|:-------:|
> |  Original   |  0.88   |   0.43   |   0.33   |  0.83   |
> |     TIP     |  0.88   |   0.47   |   0.33   |  0.82   |
> | CyclicReflex|  0.90   |   0.50   |   0.40   |  0.88   |
>
> CyclicReflex consistently outperforms both Original and TIP across all four benchmarks, even when the model size increases and the model family changes. This provides an evidence  that our method is not specific to a single lineage (DeepSeek-distilled models), but remains effective for larger models and different architectures.
>
> To note, 14B is the largest model that our computing resources can afford. We hope to add more model types in our revision if more computing resources become available to us.

---

> ### Author Response · Authors · 2025-11-20
> **2. Response on related work (Weakness 2)**
>
> We appreciate the reviewer for pointing out the missing references [1–3]; we will explicitly add them to the related-work section and discuss their relationship to our method.
>
> Conceptually and technically, these works target different aspects of test-time scaling from ours:
>
> - Kinetics [1] studies test-time scaling laws from a macroscopic perspective, focusing on how to distribute test-time compute across model sizes and attention sparsity under realistic KV-memory constraints. In contrast, our work assumes a fixed model and a fixed test-time compute budget per sample, and focuses on micro-level allocation of reflection tokens within a single reasoning trajectory via a lightweight logits-scheduling policy.
>
> - Self-Certainty [2] proposes a scoring rule for best-of-N selection, i.e., how to rank and select among multiple completed samples using a self-certainty signal. Our setting keeps the sample count $N$ fixed and instead modifies how each individual trajectory is generated by modulating reflection tokens inside that trajectory. In other words, [2] operates on cross-sample selection, whereas CyclicReflex operates on intra-trajectory token allocation.
>
> - Iterative Deepening Sampling [3] introduces a multi-round test-time scaling framework with manually triggered self-reflection and geometrically increasing budgets. This is a substantially different algorithmic paradigm compared to our single-pass cyclical scheduling: [3] repeatedly expands the sampling budget and re-invokes self-correction, while CyclicReflex only requires  one decoding pass with a periodic schedule over reflection-token logits. Moreover, to the best of our knowledge, there is currently no official open-source implementation for [3].
>
> We will add a dedicated paragraph in the revised version clarifying: (1) how [1–3] address global test-time scaling (model size, N, multi-round sampling), whereas CyclicReflex focuses on local reflection-token scheduling within a single trajectory; and (2) that CyclicReflex could in principle be combined with them (e.g., using CyclicReflex inside each sampled trajectory of a best-of-N or iterative-deepening procedure).

---

> ### Author Response · Authors · 2025-11-20
> **3. Response on pass@N and cons@N metrics (Weakness 4)**
>
> Thank you for your comments. Following your suggestion, we evaluated the consistency of our method under both pass@N and cons@N on MATH500 using DeepSeek-R1-Distill-Qwen-7B. The results are reported in **Tables R7 and R8**.
>
> Table R7. pass@N of DeepSeek-R1-Distill-Qwen-7B on MATH500.
>
> |   Method    | pass@1 | pass@2 | pass@4 | pass@8 | pass@16 |
> |-----------|:------:|:------:|:------:|:------:|:-------:|
> |  Original   |  0.74  |  0.82  |  0.88  |  0.91  |  0.93   |
> |     TIP     |  0.75  |  0.83  |  0.87  |  0.90  |  0.91   |
> | CyclicReflex|  0.77  |  0.84  |  0.89  |  0.93  |  0.95   |
>
> Table R8. cons@N of DeepSeek-R1-Distill-Qwen-7B on MATH500.
>
> |   Method    | cons@1 | cons@2 | cons@4 | cons@8 | cons@16 |
> |-----------|:------:|:------:|:------:|:------:|:-------:|
> |  Original   |  0.74  |  0.75  |  0.79  |  0.82  |  0.85   |
> |     TIP     |  0.75  |  0.76  |  0.79  |  0.82  |  0.84   |
> | CyclicReflex|  0.77  |  0.78  |  0.80  |  0.83  |  0.86   |
>
> We can observe the following:
>
> - CyclicReflex consistently outperforms both Original and TIP across all pass@N and cons@N values, confirming that our method improves standard test-time scaling metrics, not only single-sample accuracy.
> - TIP can improve over Original for small N (e.g., pass@1, pass@2), but its gains diminish or even reverse as N increases, likely because TIP aggressively suppresses reflection tokens, which reduces diversity and limits the benefit of sampling more trajectories. In contrast, the advantage of CyclicReflex stays consistent as N grows.
>
> We will incorporate these results into the revised version.

---

> ### Author Response · Authors · 2025-11-20
> **4. Response to LiveCodeBench evaluation setting (Question 1)**
>
> Thank you for raising this important question about LiveCodeBench and how reflection tokens interact with code generation.
>
> In our LiveCodeBench setup:
>
> - The model first produces a natural-language reasoning trace, delimited by special tags such as `<think> ... </think>`.
> - After the `</think>` marker, the model then produces the final answer, which may contain a mix of natural language and code. Only the code portion is extracted and executed for evaluation, following the standard LiveCodeBench protocol.
>
> As mentioned in Line 329，CyclicReflex is designed to act only on the reasoning phase, i.e., it modulates the logits of reflection tokens before `</think>`. Concretely:
>
> - We apply our cyclical logits schedule to reflection-related tokens only while the current prefix is inside the `<think> ... </think>` block.
> - After the model outputs `</think>`, CyclicReflex is disabled, and decoding proceeds with the standard logits (no further reflection-token modulation).
>
> Thus, CyclicReflex still improves performance on LiveCodeBench (see Fig. 5) by refining the reflection tokens in the reasoning trace, which in turn leads to higher-quality code in the final answer.

---

> ### Author Response · Authors · 2025-11-20
> **5. Response to hyperparameter search and random seeds (Question 2 and Question 3)**
>
> We appreciate the reviewer’s concern regarding reproducibility and the details of our hyperparameter search. We provide additional detail below.
>
> (a) How the grid search is done and whether it is per-model per-dataset
>
> As noted in Appendix A, CyclicReflex has two main hyperparameters:
>
> - the period $C$, which controls how frequently reflection tokens are promoted vs. suppressed;
> - the amplitude $A$, which controls the strength of this modulation.
>
> We conducted a grid search over $(A, C)$ to understand sensitivity. The key observations (also reflected in Fig. 4 and Fig. A1) are:
>
> - Performance is more sensitive to $C$ than to $A$.
> - At $C = 600$, the method achieves its best performance in our search. Different amplitudes $A \in \{5, 7, 9\}$ yield very similar strong performance (Fig. A1(a)), indicating relative robustness to the choice of $A$.
>
> Based on these findings, to keep the method easy to implement, in all subsequent experiments we:
>
> - Fix the amplitude to a single value: $A = 5.0$.
> - Tune only the period $C$, and do so per benchmark family, not per model. Specifically:
>   - MATH500 and AMC2023 share the same period $C$.
>   - The AIME-style benchmarks share another common period $C$.
>   - For a given benchmark, the same $C$ is used across all evaluated models.
>
> $C$ is related to the model’s generation length when answering questions. Longer generations benefit from a larger period $C$, while shorter generations favor a smaller one. This explains why MATH500, which is relatively easier and produces shorter generations, prefers a smaller $C$ than AIME. For a new task or a different reasoning-style model, we recommend setting $C \approx 0.8 \times \text{(Average generation length)}$.
>
> (b) Number of runs and averaging
>
> All reported accuracy results (e.g., those in Table 4) are averaged over 3 independent runs with different random seeds. We will explicitly state this in the main text to make the experimental protocol clearer.
>
>
> > [1] Sadhukhan R, Chen Z, Zheng H, et al. Kinetics: Rethinking Test-Time Scaling Laws. arXiv preprint arXiv:2506.05333, 2025.
> >
> > [2] Kang Z, Zhao X, Song D. Scalable Best-of-N Selection for Large Language Models via Self-Certainty. arXiv preprint arXiv:2502.18581, 2025.
> >
> > [3] Chen W, Koenig S, Dilkina B. Iterative Deepening Sampling as Efficient Test-Time Scaling. arXiv preprint arXiv:2502.05449, 2025.

---

> > ### Comment · Reviewer_Fqv6 · 2025-11-23
> >
> > I thank the authors for the rebuttal. Before I list my detailed response, I would like to remind the authors that ICLR does allow paper revision during rebuttal time, so no score will be updated from my side until the promised changes are made to the paper draft.
> >
> > Specifically, thank you for your updated results on Qwen3-14B. I understand the compute constraint, so I would like to encourage the authors to maybe also include some smaller models like Qwen3-8B or even Qwen3-4B and potentially GPT-oss-20 B to support the claim better. Also, please do include non-math dataset for these models if possible.
> >
> > For related work, please do incorporate these papers into your paper draft.
> >
> > For pass@N and cons@N results, it seems to me that under standard metrics, the proposed method only provides a small benefit when N is small, and the improvement margin is extremely limited when N becomes bigger. Thus, I would not agree with the claim of "the advantage of CyclicReflex stays consistent as N grows."
> >
> > For LivecodeBench, I understand that the approach only changes in terms of the thinking process; however, reasoning models still draft code pieces in such parts. So if I understand correctly, there is no special consideration for potential linguistic constraints in this case, is that correct?
> >
> > For hyperparameters, it seems to me that the recommended hyperparameters are not the ones that are actually used in the main results, e.g., the original rollout gives an average length of 780 on MATH500, leading to a C that should be smaller than 400, which is not the 600 that yields the best performance. Thus, it raises the problem of whether hyperparameter search is necessary for CyclixReflex to gain a benefit.

---

> > > ### Author Response · Authors · 2025-11-24
> > > **Additional response to Reviewer Fqv6.**
> > >
> > > Thank you for the reminder. We have already revised the paper and uploaded it accordingly.
> > >
> > > 1. **Response to the request for more models and non-math datasets.**
> > > We have run additional experiments on the suggested models, including Qwen3-4B, Qwen3-8B, and Qwen3-14B.  And the  non-math benchmark LiveCodeBench has been considered. The results are reported in Tables R9, R10, and R11 below. Across all three model sizes, CyclicReflex consistently improves over both the Original decoding and TIP, including the evaluation on the non-math LiveCodeBench dataset.
> > >
> > >     **Table R9. Accuracy of **Qwen3-4B** on MATH500, AIME2024, AIME2025, AMC2023, and LiveCodeBench.**
> > >
> > >     |    Method    | MATH500 | AIME2024 | AIME2025 | AMC2023 | LiveCodeBench |
> > >     |-------------|:-------:|:--------:|:--------:|:-------:|:-------------:|
> > >     | Original    |  0.60   |   0.13   |   0.03   |  0.47   |     0.58      |
> > >     | TIP         |  0.63   |   0.17   |   0.07   |  0.49   |     0.59      |
> > >     | CyclicReflex|  0.68   |   0.23   |   0.10   |  0.53   |     0.61      |
> > >
> > >     **Table R10. Accuracy of **Qwen3-8B** on MATH500, AIME2024, AIME2025, AMC2023, and LiveCodeBench.**
> > >
> > >     |    Method    | MATH500 | AIME2024 | AIME2025 | AMC2023 | LiveCodeBench |
> > >     |-------------|:-------:|:--------:|:--------:|:-------:|:-------------:|
> > >     | Original    |  0.83   |   0.37   |   0.27   |  0.79   |     0.59      |
> > >     | TIP         |  0.84   |   0.40   |   0.23   |  0.80   |     0.60      |
> > >     | CyclicReflex|  0.86   |  0.43    |   0.33   |  0.84   |     0.63      |
> > >
> > >     **Table R11. Accuracy of **Qwen3-14B** on MATH500, AIME2024, AIME2025, AMC2023, and LiveCodeBench.**
> > >
> > >     |    Method    | MATH500 | AIME2024 | AIME2025 | AMC2023 | LiveCodeBench |
> > >     |-------------|:-------:|:--------:|:--------:|:-------:|:-------------:|
> > >     | Original    |  0.88   |   0.43   |   0.33   |  0.83   |     0.63      |
> > >     | TIP         |  0.88   |   0.47   |   0.33   |  0.82   |     0.63      |
> > >     | CyclicReflex|  0.90   |   0.50   |   0.40   |  0.88   |     0.67      |
> > >
> > > 2. **Response to the related-work suggestion.**
> > > We have incorporated the suggested references into the revised paper. In particular, we added a discussion of these works in Lines 144–150 of the updated draft to more clearly position CyclicReflex relative to existing work.
> > >
> > > 3. **Response to pass@$N$ and cons@$N$ concerns.**
> > > We respectfully disagree with the interpretation that CyclicReflex only provides a “small” benefit and that the advantage does not persist as $N$ grows. Recent work on test-time scaling and RLVR has shown that, as $N$ becomes large, the performance of many advanced methods converges towards that of the original model under its base decoding strategy [1,2]. Mathematically, pass@$N$ measures the probability that *at least one* out of $N$ samples is correct. Therefore, it is not surprising that as $N$ increases, this quantity gets improved and  the absolute margin between methods gets shrinked.
> > >
> > >
> > >     In this context, we view it as encouraging that CyclicReflex still maintains a consistent advantage even at relatively large N. For example, on Qwen3-14B and MATH500, CyclicReflex improves pass@16 by 4 percentage points over TIP. We emphasize that the narrowing of absolute gaps as $N$ increases does not diminish the effectiveness or promise of our approach; this behavior is expected, and the fact that CyclicReflex continues to outperform the baselines, even with smaller margins, demonstrates that it delivers meaningful gains even with a larger budget on $N$.
> > >
> > >
> > > 4. **Response to the LiveCodeBench question.**
> > > Yes, on LiveCodeBench, CyclicReflex only alters the *natural-language reflection tokens* of the reasoning trace. However, if you had a specific type of “linguistic constraint” in mind, we would be very happy to try your suggestion further.
> > >
> > > 5. **Response to the hyperparameter concern.**
> > > We apologize for a typo in our initial response regarding the hyperparameter $C$. Our intended rule-of-thumb is
> > > $C/2 \approx 0.4 \times \text{(average generation length)}$, where the right-hand side was supposed to determine the half-radius of the period $C$, namely, $C/2$. Therefore, our suggested rule should be $C \approx 0.8 \times \text{(average generation length)}$. For MATH500, the average generation length of the original rollout is about 780 tokens, so choosing $C = 600$ is consistent with this guideline and empirically yields the best performance in our experiments.
> > >
> > > > [1] Yue, Yang, et al. "Does reinforcement learning really incentivize reasoning capacity in llms beyond the base model?." arXiv preprint arXiv:2504.13837 (2025).
> > > >
> > > > [2] Zainullina, Karina, et al. "Guided Search Strategies in Non-Serializable Environments with Applications to Software Engineering Agents." arXiv preprint arXiv:2505.13652 (2025).

---

### Official Review · Reviewer_cPcQ · 2025-10-31

**Soundness:** 3
**Presentation:** 3
**Contribution:** 3
**Rating:** 4
**Confidence:** 3

**Summary:**

This paper introduces CyclicReflex, a novel, training-free decoding strategy for Large Reasoning Models (LRMs) that dynamically modulates the logits of "reflection tokens" (e.g., "wait", "but") using a cyclical, triangular waveform. The core motivation is the "resource allocation" problem for reflection tokens, where models suffer from either under-reflection (premature convergence to an answer) or over-reflection (excessive, looping reasoning). The authors draw a conceptual analogy between scheduling reflection tokens and scheduling learning rates in optimization, where under/over-reflection mirror the effects of too small/large a learning rate. Inspired by cyclical learning rates, CyclicReflex alternates between promoting and suppressing reflection tokens, balancing exploration and convergence in the reasoning "thought landscape." Extensive experiments on math and non-math benchmarks show that CyclicReflex consistently improves accuracy over baseline decoding and competing methods like TIP and S1, without significantly increasing generation length, and integrates well with other test-time scaling techniques.

**Strengths:**

* The core idea of treating reflection tokens as a schedulable resource and drawing an analogy to learning rate scheduling is highly original and compelling.

* CyclicReflex is simple to implement, incurs no additional training cost, and demonstrates consistent, non-trivial performance gains across a wide range of tasks and model scales.

* The experiments are thorough, covering multiple datasets (MATH, AIME, AMC, GPQA, LiveCodeBench), model families (Qwen, LLaMA), and integration with other test-time scaling methods (Best-of-N, Beam Search).

**Weaknesses:**

* The method introduces two new hyperparameters (amplitude A, period C). The grid search ranges provided in Appendix A are helpful, but the paper could do more to provide heuristics or guidelines for setting these parameters on new tasks or models, as optimal values seem to vary by dataset (e.g., different C for MATH500 vs. AIME).

* The set of reflection tokens V_hat is a critical component, but its definition and potential sensitivity are not deeply discussed. Is the performance robust to the specific choice of tokens? Could the method be extended to learn or identify reflection tokens dynamically?

**Questions:**

* How would you advise a practitioner to set the hyperparameters A and C for a new, unseen task or a different LRM architecture not tested in this work? Are they transferable across similar domains?

* How was the vocabulary of reflection tokens V_hat constructed? Is the method's performance sensitive to minor changes in this set (e.g., adding "however" or "therefore")?

* The learning rate analogy is powerful, but in optimization, the loss landscape is fixed, whereas the "thought landscape" is generated on-the-fly. How does this fundamental difference impact the interpretation of your method?

---

> ### Author Response · Authors · 2025-11-20
> **1. Response on hyperparameter sensitivity and practical guidelines (Weakness 1 and Question 1)**
>
> Thank you for pointing out the need for clearer guidance on the new hyperparameters, and we apologize if our design choices were not sufficiently clear in the original text.
>
> As noted in Line 468, the key hyperparameter in CyclicReflex is the period $C$. Our ablation studies (Fig. 4 and Fig. A1) report the model performance against $C$, showing that accuracy peaks around  $C = 600$. In contrast, once $C$ is set to a good value, the method is comparatively less sensitive to the amplitude $A$: in Fig. A1(a), $A = 5$, $7$, and $9$ all produce similarly strong performance when $C = 600$.
>
> We emphasize that the above sensitivity is consistent with the theory of stepsize hedging [1,2], which emphasizes that the schedule pattern, in our case, the period of alternating between “encourage reflection” and “suppress reflection” phases, has a larger effect on convergence than the exact magnitude of each step. Translating the analogy, $C$ plays a similar role to the schedule’s temporal structure, while $A$ behaves more like a modest rescaling of the step size.
>
> After observing this pattern, for the remaining experiments we fix the amplitude to $A = 5.0$ and tune only the period $C$. $C$ is related to the model’s generation length when answering questions. Longer generations benefit from a larger period $C$, while shorter generations favor a smaller one. This explains why MATH500, which is relatively easier and produces shorter generations, prefers a smaller $C$ than AIME. For a new task or a different reasoning-style model, we recommend setting $C \approx 0.8 \times \text{(Average generation length)}$.
>
> We also find that the suggested hyperparameters generalize across domains in our experiments (Sec. 5). We used the same $(A, C)$ pair for mathematical reasoning (AIME2024, AIME2025), code generation (LiveCodeBench), and general-knowledge reasoning (GPQA Diamond), and observe consistently strong performance. This supports the view that CyclicReflex does not require heavy per-task hyperparameter tuning and remains practical as a “training-free’’ test-time method.
>
> We will add practical guidelines for choosing $C$ and $A$ in the revision to address concerns about sensitivity and usability.

---

> ### Author Response · Authors · 2025-11-20
> **2. Response on robustness to the choice of reflection tokens $\hat{V}$ (Weakness 2 and Question 2)**
>
> Thank you for raising the concern about the subjectivity of the reflection-token definition and its robustness to linguistic drift.
>
> In our main experiments, the reflection-token set is constructed by extending prior work on reflective reasoning and test-time scaling [3,4], and by analyzing which tokens are most frequently used by the model when it “thinks out loud” on a held-out validation set. The resulting set $\hat{V}$ contains frequent tokens that signal intermediate reasoning, uncertainty, or self-correction.
>
> However, the set of reflection tokens is not necessarily pre-specified or static. To directly address your concern, we further evaluate CyclicReflex using a dynamic reflection-token set rather than a fixed, pre-defined list. Concretely, during decoding we continuously expand the reflection-token set by incorporating tokens that the model itself uses in contexts strongly indicative of reflective reasoning. The procedure is as follows:
> - Initialization. Begin with a small seed set of reflection tokens (e.g., “wait”, “but”).
> - Dynamic expansion. To increase the diversity of the reflection-token set, we expand it dynamically during decoding. At decoding step $t$, if the generated Top-1 token is in the current reflection-token set and if the logit gap between Top-2 and Top-1 is smaller than the gap between Top-2 and Top-3, we add the Top-2 token to the reflection-token set.
>
> The intuition is that such a condition identifies tokens that the model tends to use in contexts highly similar to those of existing reflection tokens, thereby expanding the set in this way. We find that after tuning this dynamic expansion procedure, the resulting reflection-token set indeed becomes richer and more diverse, capturing additional reflection-like tokens such as “hmm”, “maybe”, “perhaps”, and others that do not appear in the original curated list.
>
> We evaluate CyclicReflex with both the fixed and dynamic reflection-token sets on DeepSeek-R1-Distill-Qwen-7B across MATH500, AIME2024, AIME2025, and AMC2023, and compare them to the Original decoding and TIP. The results are shown in **Table R5**.
>
> Table R5. Accuracy of DeepSeek-R1-Distill-Qwen-7B on MATH500, AIME2024, AIME2025, and AMC2023.
>
> |                  Method                   | MATH500 | AIME2024 | AIME2025 | AMC2023 |
> |----------------------------------|:-------:|:--------:|:--------:|:-------:|
> |                 Original                  |  0.86   |   0.43   |   0.31   |  0.81   |
> |                   TIP                     |  0.87   |   0.43   |   0.30   |  0.85   |
> | CyclicReflex w/ dynamic reflection token set |  0.89   |   0.50   |   0.37   |  0.91   |
> | Ours: CyclicReflex  |  0.89   |   0.50   |   0.37   |  0.90   |
>
> The performance of CyclicReflex with the fixed and dynamic reflection-token sets is similar across the datasets. This suggests that CyclicReflex is robust to the precise choice of reflection tokens and does not hinge on an empirical list. In particular, even when the model starts to use new expressions of reflection (i.e., linguistic drift), our dynamic expansion mechanism continues to capture these tokens and maintain strong performance.
>
> We will add the above dynamic variant and robustness analysis in the revision.Thanks for your great comments!

---

> ### Author Response · Authors · 2025-11-20
> **3. Response to the difference between the loss landscape and the thought landscape (Question 3)**
>
> We appreciate the reviewer’s insightful comment on the analogy between learning rate scheduling in optimization and cyclical reflection-token scheduling, and in particular the question about the difference between a fixed loss landscape and an on-the-fly “thought landscape”.
>
> We would like to clarify that, in our framework, the “thought landscape’’ is **fixed** for a given model and evaluation dataset, rather than constructed on-the-fly, analogous to how the loss landscape is defined for a model with respect to a fixed training set [5]. To construct this thought landscape, we:
>
> 1. Sample a large number of reasoning traces $r$ from the model.
> 2. For each trace, decompose it into intermediate reasoning steps $r_i$.
> 3. For each step $r_i$, compute a distance between $r_i$ and the final answer $y$. As stated around Line 286, this distance metric captures the model’s uncertainty by taking the inverse probability of generating $y$ conditioned on $r_i$, normalized by the length of $y$.
>
> Once this mapping is defined, the thought landscape is fixed in the same sense as the loss landscape in parameter space: it is a static function determined by the model and the evaluation data distribution.
>
> What is generated on-the-fly is the reasoning trajectory itself: During test-time compute, for each new query, the model produces a sequence of reasoning steps. We then project these steps onto the pre-defined thought landscape with t-SNE following [6], yielding a concrete reasoning trajectory that unfolds over this landscape surface.
>
> We will clarify this construction in the revised manuscript so that the intended analogy is better justified and the role of the thought landscape is more clearly explained.
>
> > [1] Altschuler, Jason M., and Pablo A. Parrilo. “Acceleration by stepsize hedging: Silver stepsize schedule for smooth convex optimization.” Mathematical Programming (2024): 1–14.
> >
> > [2] Altschuler, Jason M., and Pablo A. Parrilo. “Acceleration by stepsize hedging: Multi-step descent and the silver stepsize schedule.” Journal of the ACM 72.2 (2025): 1–38.
> >
> > [3] Wang, Yue, et al. “Thoughts are all over the place: On the underthinking of o1-like llms.” arXiv preprint arXiv:2501.18585 (2025).
> >
> > [4] Muennighoff, Niklas, et al. “s1: Simple test-time scaling.” Proceedings of the 2025 Conference on Empirical Methods in Natural Language Processing. 2025.
> >
> > [5] Li, Hao, et al. “Visualizing the loss landscape of neural nets.” Advances in Neural Information Processing Systems 31 (2018).
> >
> > [6] Zhou, Zhanke, et al. "Landscape of thoughts: Visualizing the reasoning process of large language models." arXiv preprint arXiv:2503.22165 (2025).

---

> ### Author Response · Authors · 2025-11-26
> **Gentle Follow-Up on Rebuttal and Revised Paper**
>
> Dear Reviewer cPcQ,
>
> A few days have passed since we submitted our rebuttal and the revised version of our paper. We are writing to kindly follow up and check whether you have any additional questions or comments, or if our responses have sufficiently addressed your concerns. We would be glad to continue the discussion during the open-review period, and we hope that our detailed revisions and clarifications help more clearly convey the quality and contributions of our work.
>
> Thank you again for your time, consideration, and engagement.
>
> Sincerely,
>
> The Authors

---

### Official Review · Reviewer_rfTz · 2025-11-01

**Soundness:** 3
**Presentation:** 4
**Contribution:** 3
**Rating:** 6
**Confidence:** 3

**Summary:**

This paper tackles reasoning failures in LRMs caused by under-thinking and over-thinking stemming from the mismanagement of reflection tokens. The key contribution is CyclicReflex, a training-free decoding strategy that modulates reflection token logits using a periodic, bidirectional triangular waveform. This approach is inspired by cyclical learning rate schedules in optimization, aiming to dynamically balance reasoning exploration and convergence. Experiments across six benchmarks show CyclicReflex consistently improves accuracy over baselines, notably enhancing performance across all difficulty levels where prior methods failed.

**Strengths:**

1. The paper is well-written and logically structured, clearly motivating the problem of under- and over-reflection and leading to the proposed solution.
2.  The paper addresses the important problem of managing LRM reasoning paths. Framing reflection token generation as a resource allocation problem and drawing an analogy to learning rate scheduling in optimization is clear and novel.
3. The proposed CyclicReflex method demonstrates strong and consistent accuracy improvements across a diverse set of models and six reasoning benchmarks. It shows gains across all problem difficulty levels, validating the effectiveness of its approach.

**Weaknesses:**

**Weaknesses:**

1.  I have some questions on experimental setup:
    - The `max_token` limit is set to 8192. It is unclear if this is sufficiently long for R1-style CoT generation, as significant truncation could prematurely cut off reasoning paths. This could potentially confound the analysis of under-reflection (cut off before completion) and over-reflection (never fully realized), thus impacting the reliability of the results.
    - The hyperparameter search space for the period $C$ seems somewhat arbitrary. The paper uses small ranges for different methods (e.g., $[200, 1000]$ and $[1000, 2000]$ for CyclicReflex and $[100, 1000]$ for TIP) without a clear justification for these specific bounds.
2. The paper's analysis suggests an "early-explore, late-converge" pattern is beneficial (e.g., the phase shift analysis in Fig. A3d shows $\phi=0$ is optimal). It would be interesting to see a simpler, non-cyclical baseline that implements this insight, such as a linear decay schedule (i.e., starting with a high logit boost for reflection tokens and linearly decaying it to a penalty). It is therefore unclear if the *cyclical* nature of CyclicReflex is the key contributor to its success, or if a simpler decay schedule would have achieved similar results.

**Questions:**

Please see Weaknesses above.

---

> ### Author Response · Authors · 2025-11-20
> **1. Response to max_token limit (Weakness 1.1)**
>
> Thank you for raising this question. The choice of `max_token = 8192` follows a common practice used in the recent reasoning evaluation work of others. For example, the work [1] sets “the maximum token length … to 8192” in Section 4.1 (Experimental Setup), and the work [2] also adopts 8192 as the standard maximum generation length (see Fig. 2).
>
> To further justify that this limit does not truncate R1-style CoT in a way that could confound our conclusions, we conducted an additional experiment using DeepSeek-R1-Distill-Qwen 7B with the original decoding strategy on MATH500, AIME2024, AIME2025, and AMC2023 under five different `max_token` values: 2048, 4096, 8192, 16384, and 32768, respectively. The results are shown in **Table R3**.
>
> Table R3. Accuracy of DeepSeek-R1-Distill-Qwen 7B with the original decoding strategy on MATH500, AIME2024, AIME2025, and AMC2023 under different `max_token` values.
>
> | Dataset  | 2048 | 4096 | 8192 | 16384 | 32768 |
> |:--------:|:----:|:----:|:----:|:-----:|:-----:|
> | MATH500  | 0.77 | 0.84 | 0.86 |  0.86 |  0.86 |
> | AIME2024 | 0.17 | 0.27 | 0.43 |  0.43 |  0.43 |
> | AIME2025 | 0.17 | 0.30 | 0.31 |  0.31 |  0.31 |
> | AMC2023  | 0.65 | 0.75 | 0.81 |  0.81 |  0.81 |
>
> As can be seen, the accuracy improves when increasing `max_token` from 2048 to 8192 but becomes essentially stable once `max_token ≥ 8192`. This plateau suggests that 8192 tokens are already sufficient to capture the vast majority of useful reasoning trajectories for these benchmarks. Therefore, while we fully agree that overly aggressive truncation could in principle distort the analysis of under- vs. over-reflection, the above empirical results indicate that our chosen limit of 8192 is adequate and does not materially affect the reliability of our conclusions.

---

> ### Author Response · Authors · 2025-11-20
> **2. Response on hyperparameter search space (Weakness 1.2)**
>
> Thank you for asking this question, and we apologize for not explaining this more clearly in the main text.
>
> First, it is important to clarify that the key hyperparameters in TIP and CyclicReflex have different semantics, so their ranges are not directly comparable:
>
> - In TIP, $T_0$ is defined as “the time window over which the adjustment is applied” (Line 191). It controls when the adjustment is active over the generation steps.
> - In CyclicReflex, $C$ denotes the period of the waveform that modulates the logits of reflection tokens. It controls how frequently we alternate between promoting and suppressing reflection tokens (see Line 325).
>
>
> Because of this semantic difference, we did not attempt to force $T_0$ and $C$ into the same numeric range. Instead, we chose ranges that are meaningful for each method’s underlying mechanism.
>
> Different from  $T_0$ and $C$ , the logit penalty/boost parameters play a more similar role across the methods TIP and CyclicReflex.
>
> - In TIP, the logit penalty $\alpha$ directly scales the adjustment applied to reflection tokens.
> - In CyclicReflex, the amplitude $A$ of the waveform determines the magnitude of the positive/negative logit modulation.
>
> For $\alpha$ in TIP and $A$ in CyclicReflex, we therefore used aligned search ranges to ensure a fair comparison, since both control the strength of the logit modification.
>
> In addition, when determining the search space for TIP, we explicitly followed the recommended ranges in the original TIP paper [1], which reports them as the best-performing region in its own experiments, and also expanded these ranges in our experiments to avoid inadvertently favoring our proposed method. In other words, the search spaces were not tuned to disadvantage TIP; rather, they were carefully chosen when comparing with our approach.
>
> We will clarify these semantic differences and the rationale for the chosen ranges in the revised version to make the comparison more transparent.

---

> ### Author Response · Authors · 2025-11-20
> **3. Response on linear decay baseline (Weakness 2)**
>
> We appreciate your insightful suggestion to test a simpler, non-cyclical baseline that embodies the “early-explore, late-converge” pattern. Following your advice, we implemented a linear decay schedule for reflection tokens on DeepSeek-R1-Distill-Qwen 7B and evaluated it on MATH500, AIME2024, AIME2025, and AMC2023.
>
> Specifically, the linear decay baseline starts with a high positive logit boost for reflection tokens at the beginning of generation and then linearly decays this boost across the decoding steps until it becomes a penalty (negative adjustment).
>
> We compare four methods: the original decoding strategy, TIP, CyclicReflex, and the suggested  linear decay schedule. The results are shown in **Table R4**:
>
> Table R4. Accuracy of DeepSeek-R1-Distill-Qwen 7B on MATH500, AIME2024, AIME2025, and AMC2023.
>
> |    Method     | MATH500 | AIME2024 | AIME2025 | AMC2023 |
> |-------------|:-------:|:--------:|:--------:|:-------:|
> |   Original    |  0.86   |   0.43   |   0.31   |  0.81   |
> |      TIP      |  0.87   |   0.43   |   0.30   |  0.85   |
> | Linear decay  |  0.87   |   0.45   |   0.33   |  0.87   |
> | Ours: CyclicReflex  |  0.89   |   0.50   |   0.37   |  0.90   |
>
> The linear decay schedule indeed yields some gains over  TIP, confirming that “early-explore, late-converge” is a useful principle. However, CyclicReflex consistently achieves the best performance across all four datasets. These results suggest that the cyclical and bidirectional modulation of reflection-token logits in CyclicReflex is not merely a cosmetic choice: the cyclic structure appears to provide additional benefits beyond what can be achieved with a single-pass monotone decay.
>
> We will add this linear decay baseline and the corresponding discussion to the revision, as it directly addresses your question and further highlights the contribution of the cyclical design.
>
> > [1] Zhang, Junyu, et al. "AlphaOne: Reasoning Models Thinking Slow and Fast at Test Time." arXiv preprint arXiv:2505.24863 (2025).
> >
> > [2] Muennighoff, Niklas, et al. "s1: Simple test-time scaling." Proceedings of the 2025 Conference on Empirical Methods in Natural Language Processing. 2025.
> >
> > [3] Wang, Yue, et al. "Thoughts are all over the place: On the underthinking of o1-like llms." arXiv preprint arXiv:2501.18585 (2025).

---

> ### Author Response · Authors · 2025-11-26
> **Gentle Follow-Up on Rebuttal and Revised Paper**
>
> Dear Reviewer rfTz,
>
> A few days have passed since we submitted our rebuttal and the revised version of our paper. We are writing to kindly follow up and check whether you have any additional questions or comments, or if our responses have sufficiently addressed your concerns. We would be glad to continue the discussion during the open-review period, and we hope that our detailed revisions and clarifications help more clearly convey the quality and contributions of our work.
>
> Thank you again for your time, consideration, and engagement.
>
> Sincerely,
>
> The Authors

---

### Official Review · Reviewer_9cUv · 2025-11-02

**Soundness:** 3
**Presentation:** 3
**Contribution:** 3
**Rating:** 6
**Confidence:** 4

**Summary:**

This paper investigates the problems of "overthinking" and "underthinking" in large reasoning models. The authors treat "reflection tokens" (e.g., "wait", "but") as a cognitive resource and propose CyclicReflex, a training-free decoding strategy. Drawing an analogy with cyclical learning rates (CLR) in optimization, CyclicReflex uses a position-dependent triangular wave function to periodically adjust the logits of these reflection tokens. The goal is to dynamically balance "exploration" (encouraging reflection) and "convergence" (suppressing reflection) during the reasoning process.

**Strengths:**

- The paper frames the problem from a novel and interesting perspective: treating reflection tokens as a computational resource and formalizing their management as a scheduling problem.

- The analogy between reflection scheduling and cyclical learning rates is creative and highly intuitive. The resulting method is simple, training-free, and adds minimal computational overhead, which makes it very attractive for practical application.

- The paper provides good analysis, including visualizations of the "landscape of thought," which help build intuition for why the method works.

**Weaknesses:**

- The definition of "reflection tokens" is subjective and relies on a pre-defined, fixed set. The method's dependency on this curated list is a potential vulnerability, as it may not capture all forms of model reflection.

- The core motivation is based on an analogy. While the "landscape of thought" metaphor is appealing, it lacks a rigorous theoretical foundation and strict validation. It is not as formally defined as the "loss landscape" it draws inspiration from, making the connection feel more intuitive than theoretical.

- The method appears sensitive to its new hyperparameters. The triangular wave in Equation 3 is controlled by an amplitude ($A$) and a period ($C$). The ablation studies suggest that performance can vary significantly based on the choice of these parameters, which implies that significant tuning may be required. This undermines the practical benefit of a "training-free" approach.

**Questions:**

- How robust is the method to linguistic drift? If the model learns to use other phrases for reflection (e.g., "let's re-examine", "on the other hand") that are not in the pre-defined set, will the method's effectiveness degrade or fail entirely?

- Were other dynamic baselines considered? For instance, a baseline that encourages reflection (e.g., the TIP method but with a positive $\alpha$) or a baseline that applies simple random noise to the reflection token logits? Comparing against these would help clarify whether the observed benefit comes specifically from the cyclical nature of the adjustment or just from any form of dynamic modulation.

---

> ### Author Response · Authors · 2025-11-20
> **1. Response on reflection-token selection and extension to the dynamic list (Weakness 1 and Question 1)**
>
> Thank you for raising the concern about the subjectivity of the reflection-token definition and its robustness to linguistic drift.
>
> In our main experiments, the reflection-token set is constructed by extending prior work on reflective reasoning and test-time scaling [1,2], and by analyzing which tokens are most frequently used by the model when it “thinks out loud” on a held-out validation set. The resulting set $\hat{V}$ contains frequent tokens that signal intermediate reasoning, uncertainty, or self-correction.
>
> However, the set of reflection tokens is not necessarily pre-specified or static. To directly address your concern, we further evaluate CyclicReflex using a dynamic reflection-token set rather than a fixed, pre-defined list. Concretely, during decoding we continuously expand the reflection-token set by incorporating tokens that the model itself uses in contexts strongly indicative of reflective reasoning. The procedure is as follows:
> - Initialization. Begin with a small seed set of reflection tokens (e.g., “wait”, “but”).
> - Dynamic expansion. To increase the diversity of the reflection-token set, we expand it dynamically during decoding. At decoding step $t$, if the generated Top-1 token is in the current reflection-token set and if the logit gap between Top-2 and Top-1 is smaller than the gap between Top-2 and Top-3, we add the Top-2 token to the reflection-token set.
>
> The intuition is that such a condition identifies tokens that the model tends to use in contexts highly similar to those of existing reflection tokens, thereby expanding the set in this way. We find that after tuning this dynamic expansion procedure, the resulting reflection-token set indeed becomes richer and more diverse, capturing additional reflection-like tokens such as “hmm”, “maybe”, “perhaps”, and others that do not appear in the original curated list.
>
> We evaluate CyclicReflex with both the fixed and dynamic reflection-token sets on DeepSeek-R1-Distill-Qwen-7B across MATH500, AIME2024, AIME2025, and AMC2023, and compare them to the Original decoding and TIP. The results are shown in **Table R1**.
>
> Table R1. Accuracy of DeepSeek-R1-Distill-Qwen-7B on MATH500, AIME2024, AIME2025, and AMC2023.
>
> |                  Method                   | MATH500 | AIME2024 | AIME2025 | AMC2023 |
> |------------------------|:-------:|:--------:|:--------:|:-------:|
> |                 Original                  |  0.86   |   0.43   |   0.31   |  0.81   |
> |                   TIP                     |  0.87   |   0.43   |   0.30   |  0.85   |
> | CyclicReflex w/ dynamic reflection token set |  0.89   |   0.50   |   0.37   |  0.91   |
> | Ours: CyclicReflex  |  0.89   |   0.50   |   0.37   |  0.90   |
>
> The performance of CyclicReflex with the fixed and dynamic reflection-token sets is similar across the datasets. This suggests that CyclicReflex is robust to the precise choice of reflection tokens and does not hinge on an empirical list. In particular, even when the model starts to use new expressions of reflection (i.e., linguistic drift), our dynamic expansion mechanism continues to capture these tokens and maintain strong performance.
>
> We will add the above dynamic variant and robustness analysis in the revision.Thanks for your great comments!

---

> ### Author Response · Authors · 2025-11-20
> **2. Response to theoretical foundations and validation (Weakness 2)**
>
> We appreciate the reviewer’s question regarding the theoretical underpinnings of the “landscape of thought’’ analogy.
>
> First, CyclicReflex is not purely motivated by an informal metaphor. Its design is grounded in a novel analogy between learning-rate scheduling in optimization and reflection-token scheduling in reasoning. We adopt a cyclic triangular schedule inspired by stepsize-hedging methods [3,4], which come with provable convergence acceleration guarantees in smooth convex optimization and have shown empirical success in deep learning (see Lines 64-73 and 308-321). In our framework, the “height’’ of the triangular wave corresponds to the strength of encouragement or suppression of reflection tokens, while the period corresponds to how often we switch between these phases.
>
> Second, we agree that, unlike classical optimization, reflection-token scheduling for reasoning does not yet have a clean, continuously differentiable objective that would enable a fully rigorous theoretical treatment. This limitation stems not from a lack of rigor in our analogy, but from the fact that the underlying optimization problem for reasoning trajectories is not yet formally characterized, and this limitation can be overcome  by application of non-smooth optimization theory to reflection-token scheduling, a topic that is beyond the scope of this submission. Developing such a formulation is itself an open research question. Even empirically, we emphasize that adopting “landscape of thought’’ as a conceptual bridge between the learning rate scheduling for optimization and the reflection token scheduling for reasoning is non-trivial and genuinely novel. Establishing a connection between the “reasoning trajectory” of a test-time computation procedure and the “optimization trajectory’’ in mathematical optimization is not merely metaphorical; it provides a principled framework for importing a well-studied family of schedules (cyclical triangular steps) into reasoning-time control. Even without a closed-form objective, the analogy offers actionable structure: it explains why alternating phases of exploration (over-reflection) and exploitation (under-reflection) can be beneficial, and it yields empirically effective scheduling rules.
>
> In summary, while a rigorous theoretical foundation might be addressed in the future by well established  non-smooth optimization theory, we believe the analogy itself constitutes a meaningful, novel, and empirically validated contribution.

---

> ### Author Response · Authors · 2025-11-20
> **3. Response to hyperparameter sensitivity (Weakness 3)**
>
> Thank you for pointing out the need for clearer guidance on the new hyperparameters, and we apologize if our design choices were not sufficiently clear in the original text.
>
> As noted in Line 468, the key hyperparameter in CyclicReflex is the period $C$. Our ablation studies (Fig. 4 and Fig. A1) report the model performance against $C$, showing that accuracy peaks around  $C = 600$. In contrast, once $C$ is set to a good value, the method is comparatively less sensitive to the amplitude $A$: in Fig. A1(a), $A = 5$, $7$, and $9$ all produce similarly strong performance when $C = 600$.
>
> We emphasize that the above sensitivity is consistent with the theory of stepsize hedging [3,4], which emphasizes that the schedule pattern, in our case, the period of alternating between “encourage reflection” and “suppress reflection” phases, has a larger effect on convergence than the exact magnitude of each step. Translating the analogy, $C$ plays a similar role to the schedule’s temporal structure, while $A$ behaves more like a modest rescaling of the step size.
>
> After observing this pattern, for the remaining experiments we fix the amplitude to $A = 5.0$ and tune only the period $C$. $C$ is related to the model’s generation length when answering questions. Longer generations benefit from a larger period $C$, while shorter generations favor a smaller one. This explains why MATH500, which is relatively easier and produces shorter generations, prefers a smaller $C$ than AIME. For a new task or a different reasoning-style model, we recommend setting $C \approx 0.8 \times \text{(Average generation length)}$.
>
> We also find that the suggested hyperparameters generalize across domains in our experiments (Sec. 5). We used the same $(A, C)$ pair for mathematical reasoning (AIME2024, AIME2025), code generation (LiveCodeBench), and general-knowledge reasoning (GPQA Diamond), and observe consistently strong performance. This supports the view that CyclicReflex does not require heavy per-task hyperparameter tuning and remains practical as a “training-free’’ test-time method.
>
> We will add practical guidelines for choosing $C$ and $A$ in the revision to address concerns about sensitivity and usability.

---

> ### Author Response · Authors · 2025-11-20
> **4. Response to alternative dynamic baselines (Question 2)**
>
> Thank you for the very helpful suggestion to explore other modulation baselines, i.e., a baseline that encourages reflection (e.g., the TIP method but with a positive ) or a baseline that applies simple random noise to the reflection token logits. Following your recommendation, we implemented and evaluated these two additional baselines on DeepSeek-R1-Distill-Qwen-7B:
>
> - TIP with a positive adjustment: a variant of TIP that consistently boosts reflection-token logits (instead of penalizing them).
> - Random adjustment: a baseline that adds zero-mean random noise to the reflection-token logits at each decoding step, without any structured schedule.
>
> We evaluate these two baselines on MATH500, AIME2024, AIME2025, and AMC2023, along with the Original decoding, TIP, and CyclicReflex. The results are summarized in **Table R2**.
>
> Table R2. Accuracy of DeepSeek-R1-Distill-Qwen-7B on MATH500, AIME2024, AIME2025, and AMC2023.
>
> | Method                         | MATH500 | AIME2024 | AIME2025 | AMC2023 |
> |--------------------------------|:-------:|:--------:|:--------:|:-------:|
> | Original                       |  0.86   |   0.43   |   0.31   |  0.81   |
> | TIP                            |  0.87   |   0.43   |   0.30   |  0.85   |
> | TIP w/ positive adjustment     |  0.85   |   0.40   |   0.33   |  0.83   |
> | random adjustment              |  0.82   |   0.41   |   0.30   |  0.79   |
> | Ours: CyclicReflex                   |  0.89   |   0.50   |   0.37   |  0.90   |
>
>
> From Table R2, we observe the following:
>
> - TIP with a positive adjustment performs similarly to, or slightly worse than, the original TIP. This is expected: both variants implement a unidirectional modulation (always encouraging or always discouraging reflection), and thus can effectively alleviate only one side of the under- vs. over-reflection spectrum at a time. This highlights the central role of the stepsize-hedging pattern in CyclicReflex proposed in our paper.
> - The random adjustment baseline degrades performance relative to the Original decoding. This indicates that simply injecting dynamic noise into reflection-token logits is not enough; unstructured modulation generally hurts reasoning quality.
> - CyclicReflex consistently achieves the best performance across all four datasets, clearly outperforming both TIP variants and the random baseline.
>
> We will include these additional baselines and discussions in the revised version to more clearly disentangle the effect of cyclical scheduling from other dynamic perturbations.
>
> > [1] Wang, Yue, et al. “Thoughts are all over the place: On the underthinking of o1-like LLMs.” arXiv preprint arXiv:2501.18585 (2025).
> >
> > [2] Muennighoff, Niklas, et al. “s1: Simple test-time scaling.” Proceedings of the 2025 Conference on Empirical Methods in Natural Language Processing. 2025.
> >
> > [3] Altschuler, Jason M., and Pablo A. Parrilo. “Acceleration by stepsize hedging: Silver stepsize schedule for smooth convex optimization.” Mathematical Programming (2024): 1–14.
> >
> > [4] Altschuler, Jason M., and Pablo A. Parrilo. “Acceleration by stepsize hedging: Multi-step descent and the silver stepsize schedule.” Journal of the ACM 72.2 (2025): 1–38.

---

> ### Author Response · Authors · 2025-11-26
> **Gentle Follow-Up on Rebuttal and Revised Paper**
>
> Dear Reviewer 9cUv,
>
> A few days have passed since we submitted our rebuttal and the revised version of our paper. We are writing to kindly follow up and check whether you have any additional questions or comments, or if our responses have sufficiently addressed your concerns. We would be glad to continue the discussion during the open-review period, and we hope that our detailed revisions and clarifications help more clearly convey the quality and contributions of our work.
>
> Thank you again for your time, consideration, and engagement.
>
> Sincerely,
>
> The Authors

---

### Author Response · Authors · 2025-11-24
**Upload the revised paper.**

We sincerely thank all the reviewers for their valuable feedback on our paper. Following your suggestions, we have made further revisions to the manuscript, with the main changes highlighted in blue. In particular, we have expanded the related-work discussion, provided clearer explanations of the hyperparameters, and conducted additional experiments, including evaluating robustness to the choice of the reflection-token set as well as adding more baselines and model variants. We look forward to your further feedback and suggestions.

---

### Meta-Review · Area_Chair_vj3X · 2025-12-14

**Summary:**

The paper proposes a cyclical reflection token scheduling which dynamically modulates the logits of reflection tokens during the thinking process. The idea is novel and the experimental results are good. In terms of practicality, the reviewers have concerns about hyperparameter selection. In my opinion, the idea is novel enough and can also be used elsewhere, e.g. for exploration during reinforcement learning, despite the concerns about practicality for test-time inference.

**Reviewer Concerns:**

Reviewer 9cUv has concerns about subjectivity of the selected tokens and about the theoretical foundations. The rebuttal provided experiments with a dynamic token set and explained that the idea is used as an analogy response is adequate. The reviewer asked about the sensitivity of hyperparameters and the rebuttal explained the ablation studies done.

Reviewer rfTz asked about max token limit and about an additional baseline. The rebuttal experimentally showed that the max token limit is sufficient and provided experiments with the additional baseline.

Reviewer cPcQ asked about sensitivity of hyperparameters and choice of tokens. The rebuttal explained ablation done on hyperparameters and provided experiments with a dynamic token set.

Reviewer Fqv6 commented that only one model used in experiments. The rebuttal provided experiments on another model.
The reviewer commented that pass@k results are not provided. The rebuttal provided some experiments with pass@k. The reviewer requested for more models and datasets, which the rebuttal provided..

**Reviewer Scores:**

Reviewer 9cUv's concerns are mostly addressed in the rebuttal, so the reviewer will either retain or increase the score.
Reviewer rfTz's concerns are mostly addressed in the rebuttal, so the reviewer will either retain or increase the score.
Reviewer cPcQ's questions were answered, so the reviewer will either retain or increase the score.
Reviewer Fqv6 commented that scores will be updated only after paper revision has been done. The revision addressed most of the concerns, so the reviewer will probably increase the score from the reject score.

---

### Decision · Program_Chairs · 2026-01-26

Accept (Poster)